# Study on connectivity of buried pipeline network considering nodes reliability under seismic action

Delong Huang[1]*, Zhongling Zong[1‡], Aiping Tang[2‡]

1 School of Civil and Ocean Engineering, Jiangsu Ocean University, Lianyungang, Jiangsu Province, China,
2 School of Civil Engineering, Harbin Institute of Technology, Harbin, Heilongjiang Province, China

‡ ZZ and AT also contributed equally to this work.
* 2022000018@jou.edu.cn

**Data Availability Statement:** All pipeline data used during the study were provided by Suzhou Industrial Park Qingyuan Huayan water Co. Ltd. direct requests for these materials may be made to the provider, as indicated in the project supported

## Abstract

Currently, the connectivity calculation of complex pipeline networks is mostly simplified or ignores the influence of nodes such as elbows and tees on the connectivity reliability of the entire network. Historical earthquake damage shows that the seismic performance of municipal buried pipelines depends on the ability of nodes and interfaces to resist deformation. The influence of node reliability on network connectivity under reciprocal loading is a key issue to be addressed. Therefore, based on the general connectivity probabilistic analysis algorithm, this paper embeds the reliability of nodes into the reliability of edges, and derives a more detailed and comprehensive on-intersecting minimum path recursive decomposition algorithm considering elbows, tees, and other nodes; then, based on the reliability calculation theory of various pipeline components, the reliability of various nodes in different soil is calculated using finite element numerical simulation; finally, the reliability of a small simple pipeline network and a large complex pipeline network are used as examples to reveal the importance of considering nodes in the connectivity calculation of pipeline network. The reliability of the network system decreases significantly after considering the nodes such as elbows and tees. The damage of one node usually causes the failure of the whole pipes of the path. The damage probability is greater in the area with dense elbow and tee nodes. In this study, all types of nodes that are more prone to damage are considered in detail in the calculation. As a result, the proposed algorithm has been improved in computational accuracy, which lays the foundation for further accurate calculation of pipeline network connectivity.

## Introduction

With the rapid development of cities, the scale of the urban municipal pipeline network has increased rapidly, as shown in Fig 1, and the annual growth rate of the water supply pipeline is about 5% in China. The problem of the safe operation of buried pipelines has become increasingly prominent, which has become an urgent need and focus in the field of public safety [1]. Underground pipelines are linear concealed projects with complex operating conditions and

in the Fund. We declare that all authors received privileged access to the data. Researchers can contact Suzhou Qingyuan Huayan water Co. Ltd through the following email and obtain these data. E-mail: wangyang_qyhy@163.com.

**Funding:** This research was supported by the Hainan Province Key R&D Program (Social Development) Project of China (No. ZDYF2022SHFZ089), and the Jiangsu Province Key R&D Program (Social Development) Project of China (No. BE2021681).

**Competing interests:** The authors have declared that no competing interests exist.

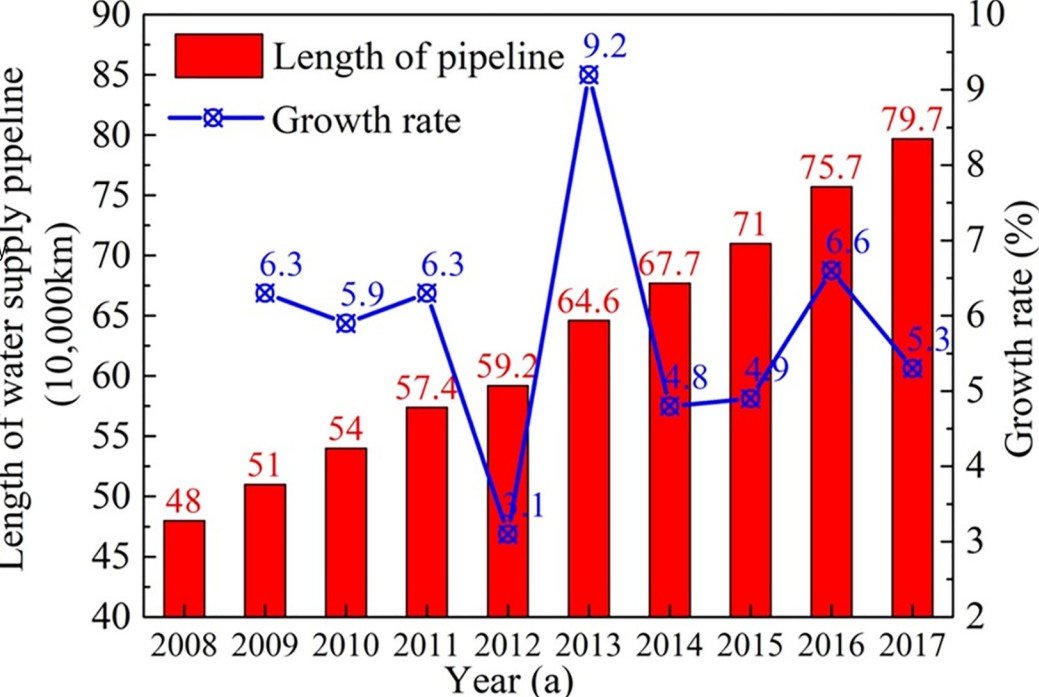

**Fig 1. Current status of water supply pipeline length in China.**

the weak parts of the direct buried pipes themselves should be strengthened to achieve the purpose of rapid evaluation and repair after earthquakes [2]. Through the investigation of the operation of water supply pipelines in various places, it is found that water supply pipes of various pipe materials and pipe diameters are prone to different degrees of damage, among which fracture, breakage, loose joints, and connection damage are the most common. After research and comparison, it can be found that the damage to buried pipelines caused by every great earthquake mainly includes [3]:

- Leakage at the interface connection of the network node. For push-on pipelines, the sockets are often disengaged, causing relative sliding between the pipelines and leakage;

- Damage occurs to the pipeline itself. It includes the longitudinal or circumferential cracks of the pipe, and the pipe passing through the fault, with a small diameter and serious corrosion, is broken, etc.;

- Damage occurs to pipeline network components. For example, fracture damage occurs at joints such as elbows, tees, and crosses. Among them, joint damage is the most common. Compared with the strength of the pipeline itself, the joints of underground pipelines and nodes such as elbows, tees, and crosses are the weak links in earthquake resistance.

The primary cause of pipeline network failure is excessive permanent ground deformation or transient ground deformation. Buried pipeline networks have the complexity of connections, intricate underground distribution, and involve large areas. The destruction of a single pipeline component often results in the pipeline connected to it being unusable. These direct and indirect factors have made buried networks severely damaged in previous great earthquakes, with huge losses [4]. Since pipeline networks are composed of pipe components, such as straight pipes, elbows, tees, etc., the analysis of the ability of the urban buried network to resist earthquakes should start with the seismic analysis of a single pipe component, and then the analysis results could be applied to the pipeline network system [5]. Whether the function

of a pipeline network system remains in normal operation, appropriate operating guidelines or standards are required to establish. It is a relatively simple operation criterion to judge the normal operation of the pipeline by whether the pipeline or components are connected. This operation criterion is adopted in the connectivity analysis of the pipeline network system. The minimum flow or minimum pressure can also be used as the standard to judge whether a pipeline works normally or provides services. This is a higher standard that is more in line with the actual situation, but it requires a large amount of calculation.

For the fragility function of a single pipeline, the "Comprehensive Research on Seismic Loss Assessment of Water Resources System" published in ASCE in 1991 [6], the relationship between various vulnerability functions and the seismic indicators used is given. These fragility functions are all based on earthquake damage investigations. It can be found that the range of the investigated earthquake damages has gradually increased, reflecting a problem that the vulnerability laws summarized through the investigation have certain limitations.

There are two levels of research content in the study of the seismic evaluation of the pipeline networks, which is the evaluation of the pipe segments leakage and the evaluation of the pipeline network reliability [7]. The research of these two parts can be called the seismic performance and overall response analysis system of the lifeline pipeline network. As for the network level, the common method at home and abroad is to use connectivity reliability to evaluate the pipeline network.

Since the network connectivity analysis is a classic "NP-Hard" problem. The number of complete minimum paths and complete minimum cuts of buried pipeline networks explodes with the scale and complexity of the pipeline network. When the system reaches a certain scale, the calculation decomposition and calculation time of the entire connectivity reliability will become extremely complicated [8]. To improve the complexity of network calculation and decomposition, many scholars have conducted in-depth research. The main achievements at this stage revolve around two basic algorithms: the probability analysis algorithm and Monte-Carlo stochastic simulation algorithm. For the Monte-Carlo algorithm, the specific calculation steps are as follows:

(1) Calculate the failure probability of every edge in the network diagram of the pipeline network system $P_{ij}$;

(2) A uniform random number $r_{ij}(0 \leq r_{ij} \leq 1)$ is generated for every edge, and the random number of every edge is compared with the failure probability $P_{ij}$ of the edge. An adjacency matrix between nodes $A$ is generated. The elements in $A$ are calculated by:

$$a_{ij} = \begin{cases} 1 & (P_{ij} < r_{ij}) \\ 0 & (P_{ij} \geq r_{ij}) \end{cases} \tag{1}$$

(3) The calculation method of the network connectivity is used to calculate the following matrix:

$$M = I + A + A^2 + \cdots + A^{n-1} \tag{2}$$

where $n$ is the total number of nodes for the above equation, and the connectivity of nodes can

be known by analyzing the elements $m_{ij}$ in the matrix $M$.

$$m_{ij} = \begin{cases} \geq 1 & \text{Connected} \\ 0 & \text{Disconnected} \end{cases} \qquad (3)$$

(4) The result calculated in the third step is recorded into the connectivity matrix $T$:

$$t_{ij} = t_{ij} + \begin{cases} 1 & (m_{ij} \geq 1) \\ 0 & (m_{ij} = 0) \end{cases} \qquad (4)$$

(5) Calculate $K$ times from (2) to (4) repeatedly to obtain the connected frequency matrix $F$ of every node (as the probability matrix).

$$f_{ij} = t_{ij}/K \qquad (5)$$

The Monte-Carlo stochastic simulation method was proposed in the 1970s and then improved by the Institute Disaster Prevention of Tongji University. The basic principle of this method is to use the frequency of the occurrence of an event to replace the probability of the event, and to conduct large-scale numerical calculation research. The connectivity reliability of the network is calculated through the Bernoulli theorem of large numbers. Zou, Chen, Liu, et al. [9–11] conducted in-depth research on pipe burst, leakage, and flow changes of the water supply network's node based on the Monte-Carlo algorithm.

For the probability analysis algorithm, researchers led by Li J. and Liu W. from Tongji University have done a lot of work. The research route of these researchers is as follows: firstly, a probabilistic analysis algorithm was proposed, which was compared with the binary decision diagram method. It is found that the recursive decomposition method has strong adaptability. The approximate value of reliability can be obtained for any complex large-scale network through this method, which has a broader development space for the actual buried engineering network [12]; then, the influence of random parameters such as pipeline stress, pipe diameter, nominal wall thickness, and pipeline yield strength is considered based on the original; and then, the seismic reliability optimization design of water supply pipe network system is proposed, and the pipeline network topology and diameter are used as the optimization parameters [13]; in recent years, Li J. et al. compared the advantages and disadvantages of genetic algorithm, genetic simulated annealing algorithm, ant colony algorithm and particle swarm optimization algorithm for the optimal design of buried pipeline network. The results show that the genetic simulated annealing algorithm performs best [14].

At present, scholars' research on the seismic connectivity of underground pipeline networks mainly focuses on the topology structure of pipeline networks and the fragility of pipelines under disaster conditions. These researches are based on the idea of probability, and they are only based on the fragility of a single straight pipeline. In this study, the connectivity of the entire pipeline network is evaluated through the idea of empirical weighting factors, and the influence of elbows and tees at nodes on the reliability of the entire pipeline network is simplified or ignored [15–22]. For a pipeline network structure with multiple nodes, consisting of multiple pipeline network components, including elbows, tees, and other special components. In addition to the complexity of the connections between pipes, the topology of the network is usually extraordinarily complex. Therefore, based on the general calculation method of pipeline network connectivity, a more detailed and comprehensive calculation method of pipeline

network connectivity considering elbows, tees, and other nodes is deduced, and a practical application is carried out. The specific research process is as follows:

- Based on the non-intersecting minimum path recursive decomposition algorithm, by correcting the reliability of the straight pipe edge pointing to the connected node, the reliability of the node is embedded in the reliability of the edge to consider the impact of node failure on the connectivity of pipeline network;

- Since the calculation of pipeline network connectivity in this paper involves the reliability of pipeline nodes, such as elbows, tees, and other pipeline network components, deduces the calculation process of the reliability of pipeline components was deduced, and the damage grade classification of pipeline and the damage limit of seismic demand was given;

- The reliability of the straight pipe, elbow, and tee node under various damage levels is calculated by the finite element method. The calculation prototype comes from the water supply pipeline in Suzhou Industrial Park, and the influence of different soil properties is considered;

- Taking a small-scale simple pipeline network as an example, the solution process of the non-intersecting minimum path recursive decomposition algorithm and the non-intersecting minimum path recursive decomposition considering nodes are explained in detail, and the results of the two methods are compared;

- Finally, the connectivity of the pipeline network is calculated by taking the large and complex water supply pipeline network in Shengpu subregion as an example, and the connectivity of the pipeline network is carried out in the cases of not-considering and considering the nodes.

## Methodology

### Non-intersecting minimum path recursive decomposition algorithm considering nodes

Considering the influence of elbow, tee, and other nodes on the system, this paper is a general weighted network, which means that the edges and nodes in the pipeline network are unreliable.

For a given system $\Omega$, a path refers to an alternating sequence of vertices and edges. A path denoted by: $\alpha = A_0 e_1 A_1 e_2 \cdots A_{n-1} e_n A_n$, where $e_i$ denotes an edge, $A_{i-1}$ and $A_i$ denote the nodes connected to $e_i$, and $A_0$ and $A_n$ denote the beginning and end of the path. If any edge is removed from the path $\mu = v_0 e_1 v_1 e_2 \cdots v_{n-1} e_n v_n$, it is impossible to form a path from $v_0$ to $v_n$, then the set of all edges in $\mu$ is called the minimum path of the system from $v_0$ to $v_n$. If the system is no longer connected after removing $n_k$ edge, then the set $\{e_{n_1}, e_{n_2}, \cdots e_{n_k}\}$ can be called the cut edge set of the system; if any edge in a cut edge set is removed and the system is restored to connectivity, then the cut edge set can be called the minimum cut.

Stipulated that the shortest minimum path $\alpha_1 = a_{11} A_{11} a_{12} A_{12} \cdots a_{1m_1} A_{1m_1}$ from the source point to the sink point in the network $\Omega$ is a reliable state, which $a_{1i}$ represents the edge in the minimum path, and $A_{1i}$ represents the elbow, tee, or other nodes connected with the edge $a_{1i}$, and the reliability of the node is embedded in the reliability of the edge pointing to the node. The minimum path $\alpha_1$ can be rewritten as $\alpha_1 = b_{11} b_{12} \cdots b_{1m_1}$, where $b_{1i}$ represents the modified reliability of the edge after embedding the reliability of the node $A_{1i}$ into the edge $a_{1i}$ pointing to the node.

Through Boolean operation and non-intersecting sum formula, the following can be obtained:

$$\Phi(\Omega) = \alpha_1 + \bar{\alpha}_1 \Phi(\Omega) \tag{6}$$

According to De Morgan's law and non-intersecting sum formula, the $\bar{\alpha}_1$ is expressed as follows [23, 24]:

$$\bar{\alpha}_1 = \bar{b}_{11} + b_{11}\bar{b}_{12} + \cdots + b_{11}b_{12} \cdots b_{1i-1}\bar{b}_{1i} + \cdots + b_{11}b_{12} \cdots \bar{b}_{1m_1} \tag{7}$$

Substituting the above equation into Eq (6), the Eq (8) can be obtained by simplification:

$$\Phi(\Omega) = \alpha_1 + \bar{b}_{11}\Phi(\Omega_{11}) + b_{11}\bar{b}_{12}\Phi(\Omega_{12}) + \cdots + b_{11}b_{12} \cdots b_{1i-1}\bar{b}_{1i}\Phi(\Omega_{1i}) + \cdots$$
$$+ b_{11}b_{12} \cdots \bar{b}_{1m_1}\Phi(\Omega_{1m_1}) \tag{8}$$

where $\Omega_{1i}$ is the network subsystem obtained after removing edges $b_{1i}$ from the network system $\Omega$.

The above-mentioned network subsystems $\Omega_{1i}(i = 1, 2,\ldots, m_1)$ can be divided into two cases, connected and non-connected subsystems, the number of which exists is $m_{1p}$ and $m_{1c} = m_1 - m_{1p}$ respectively. The former coefficient of the non-connected subsystem is a non-intersecting minimum cut. Let $\eta_{1j} = b_{11}b_{12} \cdots \bar{b}_{1j}$ be a non-connected subsystem $\Omega_{1j}$, and the corresponding non-intersecting minimum cut can be derived, i.e., the failure system function can be expressed as:

$$\Phi'(\Omega) = \sum_{j=1}^{m_{1c}} \eta_{1j} + \mu_1 \tag{9}$$

where $\mu_1$ is not all terms, i.e., $\eta_{1j}$ $(j = 1, 2,\ldots, m_{1c})$ is not a complete minimum cut.

The structure-function of the network can be written as:

$$\Phi(\Omega) = \alpha_1 + \sum_{i=1}^{m_{1p}} \beta_{1i}\Phi(\Omega_{1i}) \tag{10}$$

where $\beta_{1i} = b_{11}b_{12} \cdots \bar{b}_{1i}$, obviously, each $\beta_{1i}$ is not related to each other.

If the pipeline network is recursively decomposed by using the reliable shortest and minimum path, absorption and merger are experienced after above period, and then repeated decomposition is carried out until the decomposition is complete, the following can be obtained:

$$\Phi(\Omega) = \alpha_1 + \sum_{i=1}^{m_{1p}} \beta_i\alpha_i + \sum_{i=m_{1p}+1}^{m_{2p}} \beta_i\alpha_i + \cdots + \sum_{i=m_{(n-1)}+1}^{m_{np}} \beta_i\alpha_i \tag{11}$$

where $\beta_i$ is the coefficient after absorption and merger; $\alpha_i$ is minimum path of the $i$-th connected subsystem after passing through the whole arrangement sequence; $m_{np}+1$ is the number of all connected subsystems in the network.

Let $\omega_1 = \alpha_1$, $\omega_{i+1} = \beta_i\alpha_i$, the Eq (11) can be written as:

$$\Phi(\Omega) = \sum_{i=1}^{M} \omega_i \tag{12}$$

where $\omega_i$ is the minimum path of the pipeline network after absorption; $M = m_{np}+1$.

Then, the minimum cut accumulated for the network after decomposition is:

$$\Phi'(\Omega) = \sum_{j=1}^{K} \eta_j \tag{13}$$

where $\eta_j$ is the system's non-intersecting minimum cut; $K$ is number of the non-intersecting minimum cuts.

After the complete non-intersecting minimum path is obtained, the reliability of the pipeline network is as follows:

$$R = P[\Phi(\Omega)] = \sum_{i=1}^{M} P(\omega_i) \tag{14}$$

Embedding the reliability of nodes into the reliability of edge pointing to these nodes:

$$R = P[\Phi(\Omega)] = \sum_{b=1}^{M} \left[ \prod_{j=1}^{n} P(\omega_{b,j}) \right] \tag{15}$$

And the corrected reliability of the edge pointing to the elbow node is as follows:

$$p(\omega_{b,j}) = E_{b,j} \left( \prod_{i=1}^{e'} \bar{a}_i \right) \prod_{i=e'+1}^{1} a_i \tag{16}$$

where $E_{b,j}$ is the reliability of the elbow.

The corrected reliability of the edge pointing to the tee node is as follows:

(1) Split-flow tee

$$p(\omega_{b,j}) = T_{b,j} \left( \prod_{i=1}^{t'} \bar{a}_i \right) \prod_{i=t'+1}^{1} a_i \tag{17}$$

where $T_{b,j}$ is the reliability of the tee.

(2) Converging flow tee

$$p(\omega_{b,j}) = T_{b,j} \left( \prod_{i=1}^{t''} \bar{a}_i \right) \prod_{i=t''+1}^{2} a_i \tag{18}$$

The modified reliability of edges pointing to other types of nodes is as follows:

$$p(\omega_{b,j}) = O_{b,j} \left( \prod_{i=1}^{o'} \bar{a}_i \right) \prod_{i=o'+1}^{o'+o} a_i \tag{19}$$

where $O_{b,j}$ is the reliability of other types of nodes.

Based on the decomposed minimum path $\omega_i$, if the reliability of the node $j$ and edge is known, the reliability of the node $j$ can be put into the reliability of the edge pointing to this

node, as shown in the following equation, as follows:

$$p(\omega_{b,j}) = O_{b,j}\prod_{i=1}^{o} a_i \tag{20}$$

In the non-intersecting minimum path $\omega_i$, if the correction reliability of all the edges pointing to the node $j$ is equal to zero, there are two cases at this time. The first case is that the node $j$ fails, and the second case is that all the edges pointing to the node $j$ are invalid, as shown below:

$$p(\omega_{b,j}) = (1 - O_{b,j}) + O_{b,j}\prod_{i=1}^{o'} \bar{a}_i \tag{21}$$

The reliability of the network system $\Omega$ considering elbow, tee, and other types of nodes as:

$$R = P[\Phi(\Omega)] = \sum_{b=1}^{M}\left\{\prod_{j=1}^{n_1}\left[E_{b,j}\left(\prod_{i=1}^{e'}\bar{a}_i\right)\prod_{i=e'+1}^{1}a_i\right]\cdot\prod_{j=1}^{n_2}\left[T_{b,j}\left(\prod_{i=1}^{t'}\bar{a}_i\right)\prod_{i=t'+1}^{1}a_i\right]\cdot\prod_{j=1}^{n_3}\left[T_{b,j}\left(\prod_{i=1}^{t''}\bar{a}_i\right)\prod_{i=t''+1}^{2}a_i\right]\cdot\prod_{j=1}^{n_4}\left[O_{b,j}\left(\prod_{i=1}^{o'}\bar{a}_i\right)\prod_{i=o'+1}^{o'+o}a_i\right]\cdot\left(\prod_{i=1}^{k'}\bar{a}_i\right)\prod_{i=k'+1}^{k'+k}a_i\right\} \tag{22}$$

The failure probability of the network $\Omega$ is:

$$F = P[\Phi'(\Omega)] = \sum_{j=1}^{K} P(\eta_j) = \sum_{c=1}^{K_s}\left[\prod_{t=1}^{n'} P(\eta_{c,t})\right] \tag{23}$$

For large and complex water supply pipeline network systems, the maximum and minimum values of the connectivity reliability of the pipeline network can be calculated based on the incomplete minimum path and minimum cut, as shown in the following equation:

$$\sum_{b=1}^{M_s}\left[\prod_{j=1}^{n} P(\omega_{b,j})\right] \le R \le 1 - \sum_{c=1}^{K_s}\left[\prod_{t=1}^{n'} P(\eta_{c,t})\right] \tag{24}$$

where $M_f$ is the number of minimum paths in the water supply network that satisfies the reliability calculation accuracy, $M_s \le M$; $K_s$ is the number of minimum cuts in the water supply network that satisfies the reliability calculation accuracy, $K_s \le K$.

When the upper and lower bounds of the connectivity reliability of the network system obtained by Eq (24) reach the preset accuracy, the approximate value of the reliability can also be calculated by the following equation:

$$R_{sys} \approx \frac{1}{2}\left[1 - \sum_{j=1}^{K_s}\left[\prod_{t=1}^{n'} P(\eta_{c,t})\right] + \sum_{b=1}^{M_s}\left[\prod_{j=1}^{n} P(\omega_{b,j})\right]\right] \tag{25}$$

Through the above probability inequality, the actual engineering calculation should achieve the coordination of calculation time cost and calculation accuracy. The algorithm considering the elbow, tee, and other nodes has a larger amount of calculation, that is, the calculation efficiency is reduced, but the calculation accuracy of connectivity is improved.

## Reliability calculation theory of the pipe section and node

For one single component, seismic fragility can be expressed as [25]:

$$P_{In} = P[D > C|IM] \tag{26}$$

where $IM$ is the ground motion intensity index; $C$ is the seismic capacity threshold of the

**Table 1. Seismic damage classification of the water supply pipeline based on the leakage rate.**

| Leakage rate | $\leq 0.05$ | (0.05, 0.15] | (0.15, 0.60] | >0.60 |
|---|---|---|---|---|
| Damage state | Low | Moderate | Extensive | Complete |

pipeline network components; $D$ is seismic capacity requirements of pipeline network components.

Assuming that $C$ and $D$ are the normal distributed correlation quantities in the logarithmic coordinate system, then the Eq (26) can be transformed into [26, 27]:

$$P_{\text{In}} = P[D > C|IM] = \Phi\left[\frac{In(\hat{D}/\hat{C})}{\sqrt{\beta^2_{\text{D|IM)}} + \beta^2_{\text{C}}}}\right] \quad (27)$$

where $\hat{D}$ is the mean value of seismic requirements of network components under seismic action; $\hat{C}$ is the mean value of seismic capacity of network components under seismic action; $\beta_{\text{D|IM}}$ is the standard deviation of demand in logarithmic coordinates; $\beta_{\text{C}}$ is the standard deviation of seismic capacity in logarithmic coordinates.

According to the research of scholars, the relationship between $\hat{D}$ and $IM$ is expressed as follows [28]:

$$\hat{D} = aIM^{\text{b}} \quad (28)$$

where $a$ and $b$ is the constant obtained by fitting.

The reliability of pipeline components can be expressed by transforming Eq (28) into logarithmic space and substituting it into Eq (27):

$$P_{\text{In}} = \Phi\left[\frac{b\ln IM + \ln a - \ln\hat{C}}{\sqrt{\beta^2_{\text{D|IM}} + \beta^2_{\text{C}}}}\right] \quad (29)$$

Based on the probabilistic seismic demand analysis of components, the reliability of a component of the network can be calculated and obtained using Eq (29).

The leakage rate of the water supply network under different damage levels is shown in Table 1. The work performed by Eskandari et al. [29] was based on Monte Carlo simulation, where all operations were repeated ten thousand times, and the results of each pipe component were averaged. Finally, the leakage, fracture, leakage rate, fracture rate, and the final failure probability of components were determined. The failure state can be determined based on the leakage rate output of the component.

In this study, the development of strain is used to predict the damage degree of the pipeline leakage. The theoretical value of ultimate strain at the position where compression buckling occurs is 0.6($t/R$). According to Hall and Newmark [30] experiments on thin-walled cylinders, compression wrinkling usually begins with a strain with the following value:

$$0.15(t/R) \leq \varepsilon_{cr} \leq 0.20(t/R) \quad (30)$$

where $t$ and $R$ are the wall thickness and radius of the pipe, respectively.

The strain range of local buckling of pipes and various nodes is [0.15($t/R$), 0.6($t/R$)]. According to Table 1, the range of low damage to the pipe is [0.15($t/R$), 0.17($t/R$)]; the range of moderate damage is (0.17($t/R$), 0.22($t/R$)]; the range of extensive damage is (0.22($t/R$), 0.42($t/R$)]; the range of complete damage is (0.42($t/R$), 0.6($t/R$)].

## Calculation of reliability of pipes and nodes: Water supply and drainage system in Suzhou industrial park

Suzhou Industrial Park is affiliated to Suzhou City, Jiangsu Province, China. The planned total area of the park is 278 square kilometers as an international cooperation demonstration area. The buried water supply and drainage pipeline network in the park is managed and operated by Suzhou Industrial Park Qingyuan Huayan Water Co. LTD. There are two waterworks in the park, with a water supply scale of 650,000 m³/d. The length of the pipeline above DN300 of the water supply network is about 1000 km. The whole water supply network system is equipped with online monitoring equipment, including 15 water quality monitoring points, 70 water pressure monitoring points, and 54 flow monitoring points. There are two sewage treatment plants in the park, with a treatment scale of 350000 m³/d. There are 44 pumping stations and about 785km of pipelines in the sewage pipeline network. At present, the company has built an intelligent pipeline network GIS system, a pipeline network SCADA system, and an automatic control system for water plants and sewage pumping stations. In this study, the fragility of some pipe sections of this pipeline network is calculated. Since the environmental parameters of the pipe section in this area are basically the same, the calculation results are applied to the calculation of the connectivity of the whole pipeline network, as shown in Fig 2.

A three-dimensional finite element model of the prototype pipes was developed in ABA-QUS. The prototype shape and size of the pipes were employed to construct the finite element model, as shown in Fig 3. Half of the prototype pipelines are located in sand and the other half

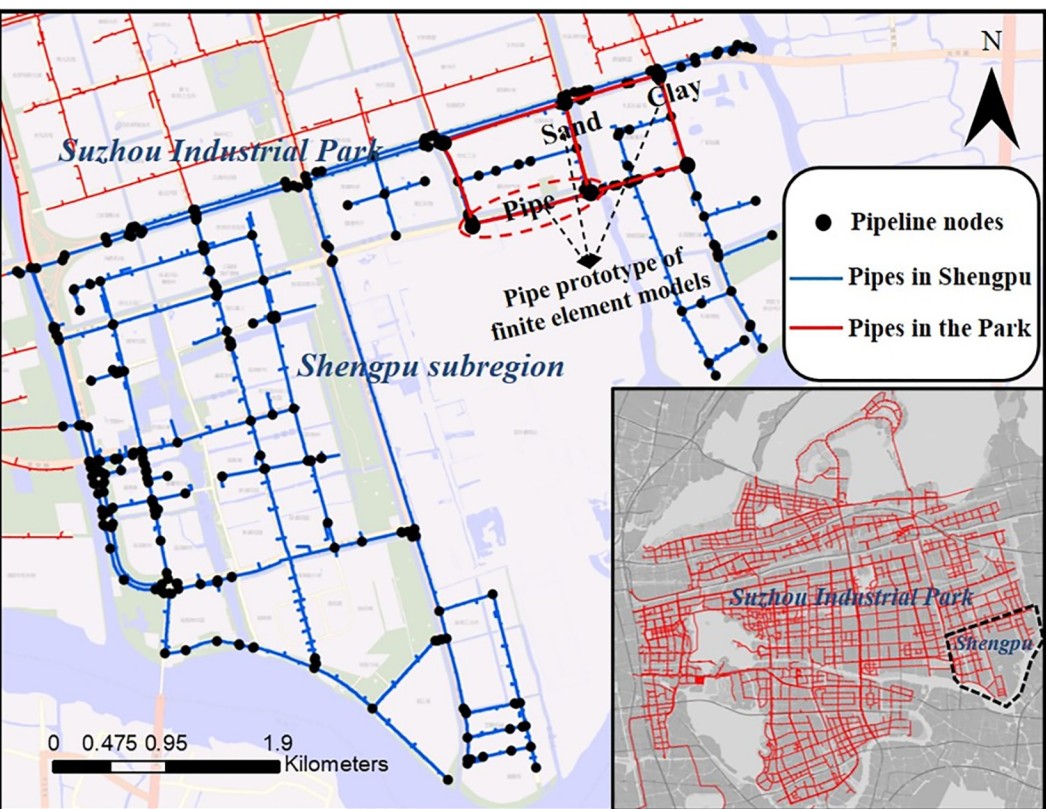

**Fig 2. Prototype pipes from the Shengpu subregion of Suzhou Industrial Park.** (Republished under a CC BY license, with permission from Suzhou Industrial Park Qingyuan Huayan Water Co. LTD, original copyright 2019).

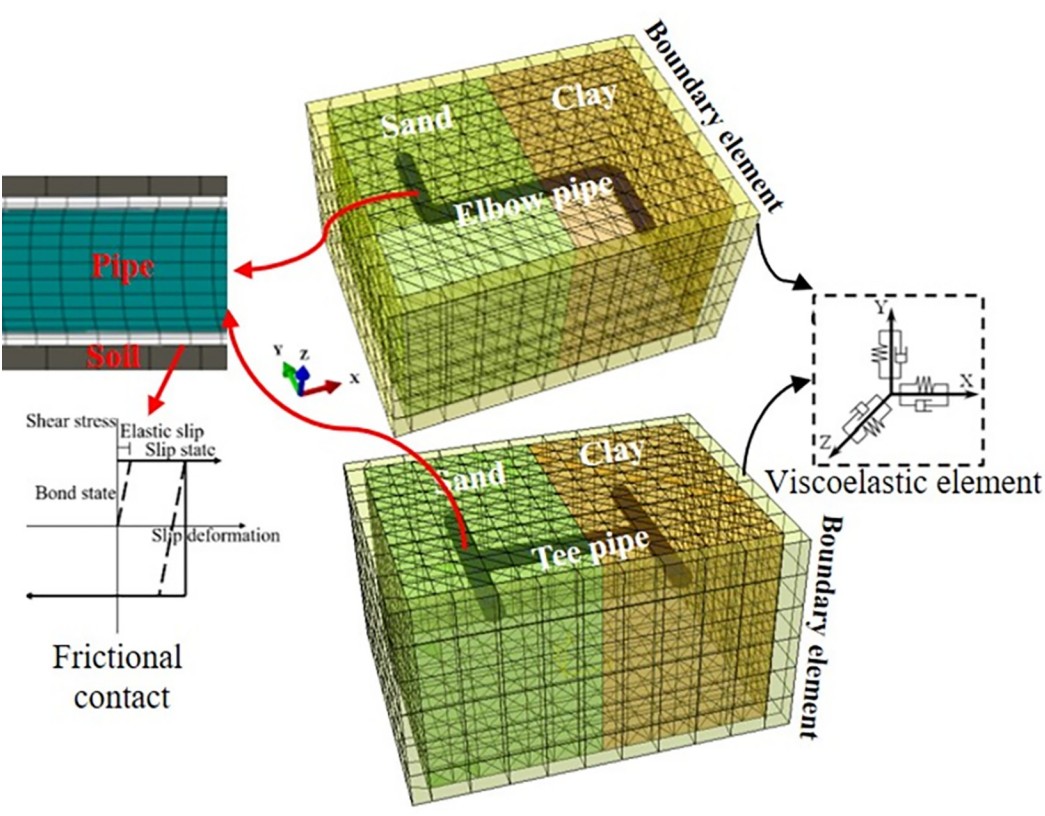

**Fig 3. Finite element models of the pipes.**

in clay. The sand and clay samples were taken, and the physical parameters of the soil were tested using a ring knife, dryer, pycnometer, and triaxial apparatus. Table 2 presents the test results. The dynamic shear modulus, shear strain, and damping of the two soil samples were measured using a dynamic triaxial instrument. Fig 4 shows the relationship between the shear strain and the normalized dynamic shear modulus and damping. The shear modulus of sand was greater than that of clay for the same shear strain, and the damping values showed the opposite trend.

To ensure the universality of the input seismic waves, 30 ground motion records of Class II and Class III sites were selected from the ground motion database of the Pacific Earthquake Engineering Center. The selected seismic records are representative in terms of duration, intensity, and frequency spectrum, and are widely distributed in magnitude, epicentral distance, and duration. Based on the nonlinear explicit dynamic calculation of finite elements [31–33], the reliability of the straight pipe, elbow, and tee under different failure levels in different soils (sandy soil, clay, and mixed soil) is obtained by using 30 groups of seismic records and pipeline network samples [34]. Fig 5 shows the failure probability of leakage of various pipe components. The fragility curves of various pipe components under four damage levels

**Table 2. Basic physical parameters of sand and clay.**

| Physical parameters | Density $\rho$ (kg/m³) | Moisture content $\omega$ (%) | Specific gravity $G_s$ | Cohesion $c$ (kPa) | Internal friction angle $\varphi$ (°) |
|---|---|---|---|---|---|
| Sand | 1792 | 0.18 | 2.62 | 0 | 32 |
| Clay | 1630 | 10.65 | 2.45 | 5 | 20 |

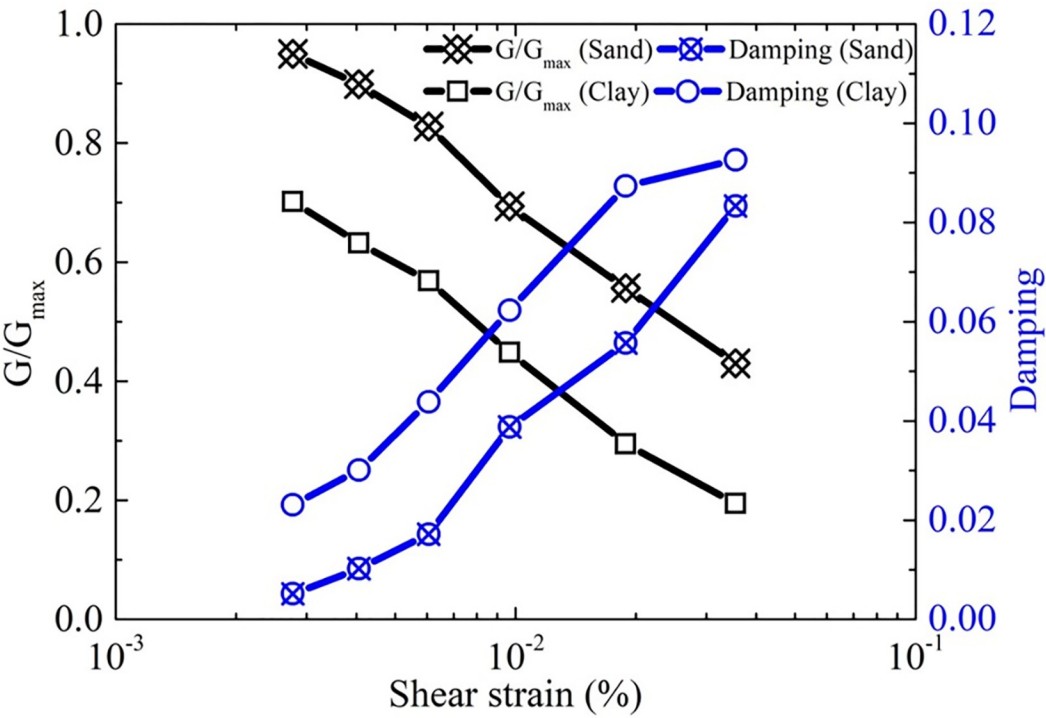

**Fig 4. Dynamic parameters of sand and clay.**

was shown in Fig 6. Table 3 shows the reliability calculation results of various types of pipes and nodes loaded with PGA = 0.2$g$ (S1–S4 Tables).

## Calculation example: Small pipeline network

### Pipeline network condition

A pipeline network with a simple topology is used to explain the solution process of the minimum path recursive decomposition algorithm and the minimum path recursive decomposition algorithm considering nodes in this example. Assume a bridge network diagram of a network system, as shown in Fig 7, where the source point is the point $s$ and the sink point is the point $t$. When calculating the connectivity, the overall reliability of the straight pipe connected to the elbow can be calculated, i.e., the reliability of the whole $2^a2^b$ and $4^a4^b$ can be calculated first, and then $2^a2^b$ and $4^a4^b$ are both represented with one number respectively. $2^a2^b$ is represented by the number 2, and $4^a4^b$ is represented by the number 4. The advantage is that the number of nodes in the pipeline network can be significantly reduced and the calculation amount can be also greatly reduced.

### Calculation results

There are a total of four minimum paths in the simple pipeline network, which are represented as $14^a4^b$, $2^a2^b5$, 135, and $2^a2^b34^a4^b$, where $2^a2^b$ and $4^a4^b$ both represent the two straight pipes connected to the elbow. Then the four minimum paths of this pipeline network can be simply represented as: 14, 25, 135, and 234. Through the above structure-function recursive decomposition algorithm, 5 non-intersecting minimum paths, and 6 non-intersecting minimum cuts can be decomposed, as shown in Fig 8.

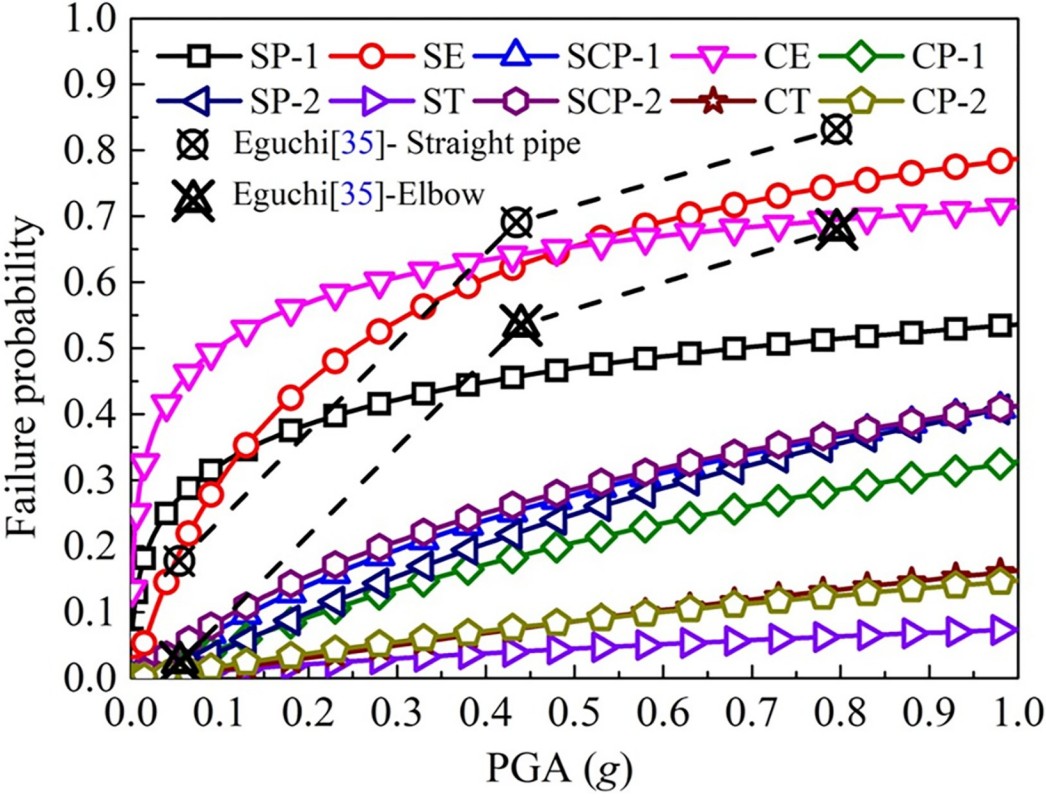

**Fig 5. Probability of leakage failure of pipe components [35].** Note: The elbows in sand and clay are represented by SE and CE respectively; the straight pipes connected to the elbow in sand and clay are represented by SP-1 and CP-1 respectively; the straight pipe connected to the elbows in the mixed soil is represented by SCP-1; the tees in sand and clay are represented by ST and CT respectively; the straight pipe connected to the tee in sand and clay is represented by SP-2 and CP-2 respectively; the straight pipe connected to the tees in mixed soil is represented by SCP-2.

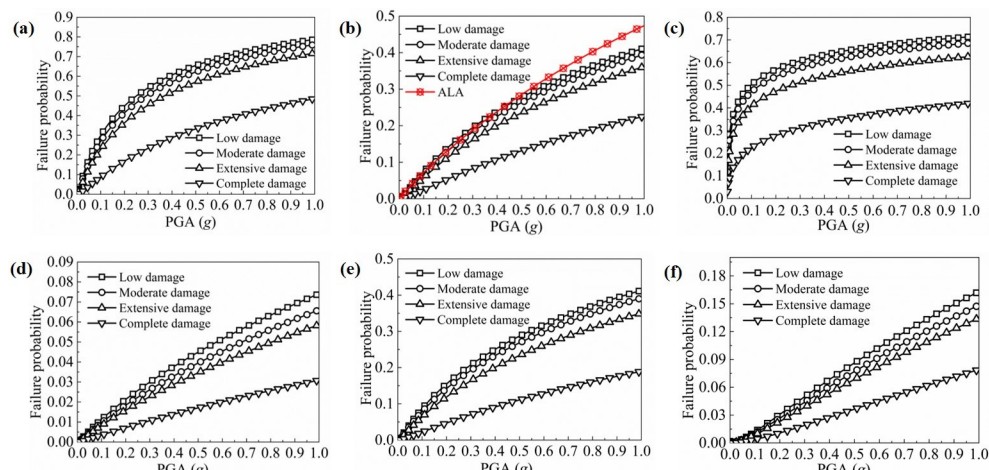

**Fig 6. Fragility of various pipe components in different soil.** (a) SE, (b) SCP-1, (c) CE, (d) ST, (d) SCP-2, (f) CT.

**Table 3. Probability of low, moderate, extensive, and complete damage to various pipe components (PGA = 0.2$g$).**

| Node type | SE | CE | SP-1 | CP-1 | SCP-1 | ST | CT | SP-2 | CP-2 | SCP-2 |
|---|---|---|---|---|---|---|---|---|---|---|
| Low | 0.449 | 0.570 | 0.385 | 0.093 | 0.141 | 0.021 | 0.031 | 0.099 | 0.037 | 0.155 |
| Moderate | 0.414 | 0.538 | 0.365 | 0.087 | 0.131 | 0.018 | 0.027 | 0.087 | 0.032 | 0.142 |
| Extensive | 0.363 | 0.473 | 0.333 | 0.077 | 0.113 | 0.016 | 0.023 | 0.066 | 0.024 | 0.119 |
| Complete | 0.168 | 0.277 | 0.195 | 0.042 | 0.054 | 0.007 | 0.011 | 0.015 | 0.005 | 0.047 |

First, $1\bar{4}$ is used for the minimum path recursive decomposition calculation in this example, and the following formula can be obtained:

$$\Phi(S) = 14 + \bar{1}\bar{4}\Phi(S) = 14 + \bar{1}\Phi(S) + 1\bar{4}\Phi(S)$$
$$= 14 + \bar{1}\Phi(S_{11}) + 1\bar{4}\Phi(S_{12}) \tag{31}$$

The system sub-network $S_{11}$ is a sub-network derived from the four minimum paths of the system $S$, in which the network edge 1 value is 0. Its structure function can be expressed as:

$$\Phi(S_{11}) = 25 \cup 234 \tag{32}$$

The system sub-network $S_{12}$ is a sub-network derived from the four minimum paths of the system $S$, in which the network edges 1 and 4 values are 1 and 0, respectively. Its structure function can be expressed as:

$$\Phi(S_{12}) = 25 \cup 35 \tag{33}$$

Then decomposing $S_{11}$ and $S_{12}$ through the minimum path event 25 of system sub-networks $S_{11}$ and $S_{12}$, respectively, then $\Phi(S)$ can be expressed as:

$$\Phi(S) = 14 + \bar{1}25 + \bar{1}\bar{2}\Phi(S_{11}) + \bar{1}2\bar{5}\Phi(S_{11}) + 1\bar{4}25 + 1\bar{4}\bar{2}\Phi(S_{12}) + 1\bar{4}2\bar{5}\Phi(S_{12})$$
$$= 14 + \bar{1}25 + 1\bar{4}25 + \bar{1}\bar{2}\Phi(S_{211}) + \bar{1}2\bar{5}\Phi(S_{212}) + 1\bar{4}\bar{2}\Phi(S_{221}) + 1\bar{4}2\bar{5}\Phi(S_{222}) \tag{34}$$

The system sub-network $S_{211}$ is the sub-network obtained by the network edge 2 value of 0 in the network system $S_{11}$, and its structure-function can be expressed as:

$$\Phi(S_{211}) = 0 \tag{35}$$

Because $S_{211}$ is a complete failure system, the event $\bar{1}\bar{2}$ becomes one of the non-intersecting minimum cut events of the network system.

System sub-network $S_{212}$ is the system sub-network resulting from the value of network edges 2 and 5 that constitute the network system $S_{11}$ are 1 and 0, respectively. Its structure

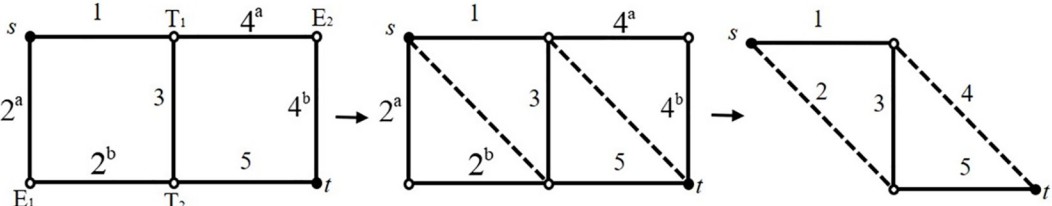

**Fig 7. Bridge network diagram and simplified form of a pipeline network.**

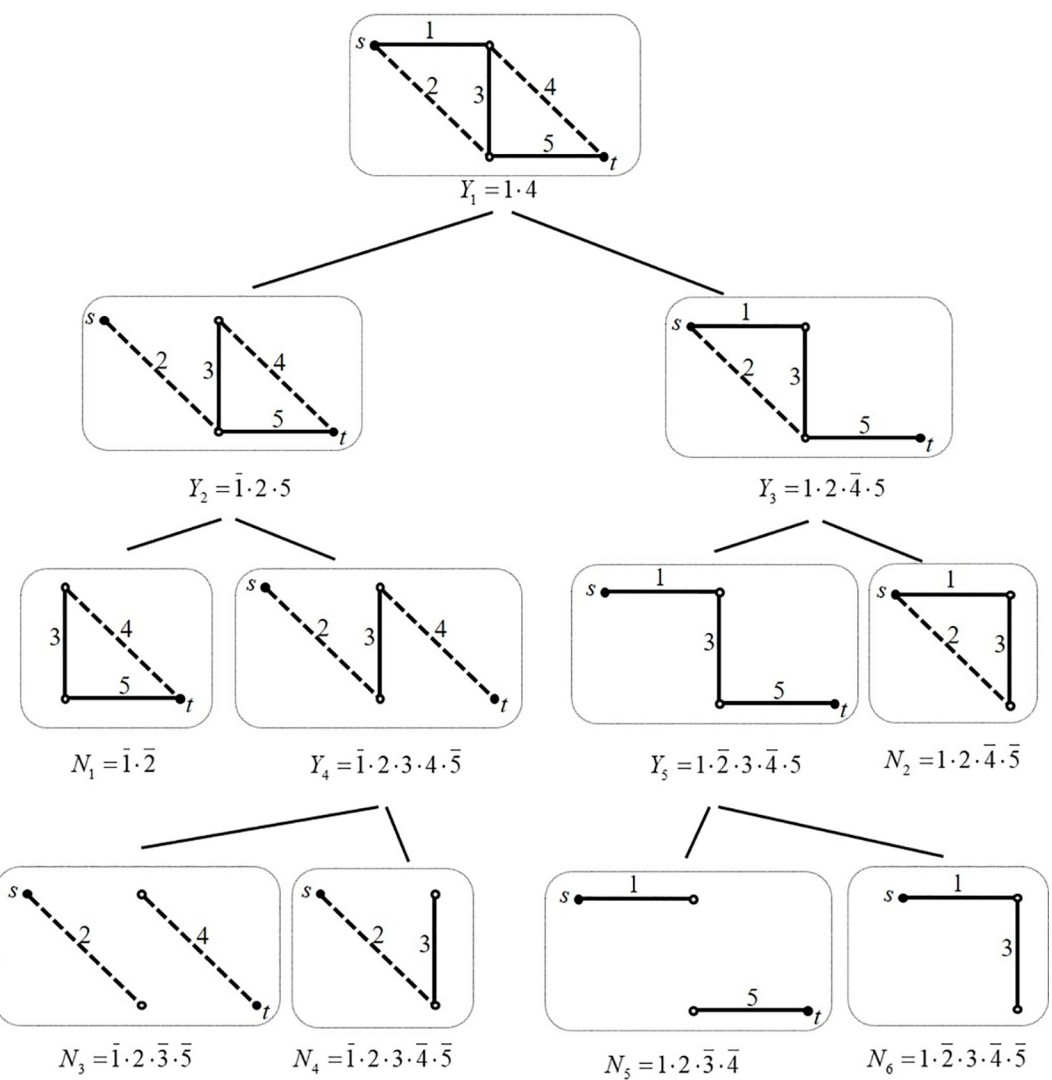

**Fig 8. The network decomposition process of the minimum path recursive decomposition algorithm.**

function can be expressed as:

$$\Phi(S_{212}) = 34 \tag{36}$$

System sub-network $S_{221}$ is the sub-network which the value of network edge 2 of the network system $S_{12}$ is 0, and the following is obtained:

$$\Phi(S_{221}) = 35 \tag{37}$$

System sub-network $S_{222}$ is the system sub-network that the value of network edges 2 and 5 that constitute the network system $S_{12}$ are 1 and 0, respectively. Its structure function can be expressed as:

$$\Phi(S_{222}) = 0 \tag{38}$$

Because $S_{222}$ constitutes a complete failure system, the event $1\bar{4}2\bar{5}$ becomes one of the non-intersecting minimum cuts events in the network system.

After the above decomposition, the structure-function and complementary structure-function of this network system can be expressed as:

$$\Phi(S) = 14 + \bar{1}25 + 1\bar{4}25 + \bar{1}2\bar{5}\Phi(S_{212}) + 1\bar{4}\bar{2}\Phi(S_{221}) \tag{39}$$

$$\Phi'(S) = \bar{1}\bar{2} + 1\bar{4}2\bar{5} + Q_1 \tag{40}$$

Then proceeding to the next step of decomposition, using the minimum path events 34 and 35 for $S_{212}$ and $S_{221}$ respectively, $S_{212}$ and $S_{221}$ are recursively decomposed again, then $\Phi(S)$ can be expressed as:

$$\Phi(S) = 14 + \bar{1}25 + 1\bar{4}25 + \bar{1}2\bar{5}34 + \bar{1}2\bar{5}\bar{3}\Phi(S_{3121}) + \bar{1}2\bar{5}3\bar{4}\Phi(S_{3122})$$
$$+ 1\bar{4}\bar{2}35 + 1\bar{4}\bar{2}\bar{3}\Phi(S_{3211}) + 1\bar{4}\bar{2}3\bar{5}\Phi(S_{3212}) \tag{41}$$

The study shows that the above system sub-networks $S_{3121}$, $S_{3122}$, $S_{3211}$ and $S_{3212}$ are all completely failed systems, so the coefficients of their structure functions constitute the non-intersecting minimum cut set of the network system accordingly. The structure-function and complementary structure-function of the network system in this example can be expressed as:

$$\Phi(S) = 14 + \bar{1}25 + 12\bar{4}\bar{5} + \bar{1}234\bar{5} + 1\bar{2}3\bar{4}5 \tag{42}$$

$$\Phi'(S) = \bar{1}\bar{2} + 12\bar{4}\bar{5} + \bar{1}2\bar{3}\bar{5} + \bar{1}23\bar{4}\bar{5} + 1\bar{2}3\bar{4} + 1\bar{2}3\bar{4}\bar{5} \tag{43}$$

Then the complete non-intersecting minimum path event of the network system can be derived. Moreover, in the above idea of recursive decomposition, the approximate value of the reliability of the network system can be obtained through the idea of the upper and lower boundary of the reliability.

The above decomposition process is aimed at the edge weight network system, i.e., the node failure is not considered. Assuming that the reliability of all network edges is set to 0.9. For convenience, the reliability of the following special network edges 2 and 4 are also set to 0.9. By using the structure function recursive decomposition, the calculation result of the network system reliability in this example is as follows:

$$R = P[\Phi(S)] = \sum_{i=1}^{5} P(\omega_i)$$
$$= P(1 \cdot 4) + P(\bar{1} \cdot 2 \cdot 5) + P(1 \cdot 2 \cdot \bar{4} \cdot 5) + P(\bar{1} \cdot 2 \cdot 3 \cdot 4 \cdot \bar{5}) + P(1 \cdot \bar{2} \cdot 3 \cdot \bar{4} \cdot 5) \tag{44}$$
$$= 0.97848$$

Similarly, the non-intersecting minimum cut method is used to calculate the reliability of the network system based on the complementary structure function, as follows:

$$R = 1 - \sum_{j=1}^{6} P(\eta_j)$$
$$= 1 - [P(\bar{1} \cdot \bar{2}) + P(1 \cdot 2 \cdot \bar{4} \cdot \bar{5}) + P(\bar{1} \cdot 2 \cdot \bar{3} \cdot \bar{5}) + P(\bar{1} \cdot 2 \cdot 3 \cdot \bar{4} \cdot \bar{5})] + P(1 \cdot \bar{2} \cdot 3 \cdot \bar{4}) + P(1 \cdot \bar{2} \cdot 3 \cdot \bar{4} \cdot \bar{5}) \tag{45}$$
$$= 0.97848$$

It can be found that the calculation results of the network reliability by using the methods of the non-intersecting minimum path and the non-intersecting minimum cut are consistent.

**Fig 9. Calculation results of simple pipeline network connectivity in example.** (a) Not considering the nodes, (b) Considering the nodes.

It is verified that if the network topology is complex, the probability inequality method can be used to obtain the partial minimum path set and partial minimum cut set of the network system, which can calculate the network reliability with predefined accuracy.

For the minimum path recursive decomposition algorithm considering nodes, the modified reliability of the edges considering the nodes should be calculated firstly. And then the connectivity probability is calculated according to the composition of the network edges with the minimum path and the minimum cut. Assuming the reliability of the elbow node, tee node and other types of nodes in the above example is 0.8, 0.7, and 0.6, respectively. Then the reliability of the network system is calculated by using the non-intersecting minimum path method considering nodes, where the $1^*$, $2^*$ and $3^*$ represent the corresponding network edges after considering the nodes, as follows:

$$
\begin{aligned}
R = P(S) &= \sum_{b=1}^{M}[\prod_{j=1}^{n} P(\omega_{b,j})] \\
&= P(1^* \cdot 4) + P(\bar{1}^* \cdot 2^* \cdot 5) + P(1^* \cdot 2^* \cdot \bar{4} \cdot 5) + P(\bar{1}^* \cdot 2^* \cdot 3^* \cdot 4 \cdot \bar{5}) + P(1^* \cdot \bar{2}^* \cdot 3^* \cdot \bar{4} \cdot 5) \\
&= 0.7972
\end{aligned}
\tag{46}
$$

Similarly, the non-intersecting minimum cut method is used again to calculate the reliability of the network system based on the complementary structure function, as follows:

$$
\begin{aligned}
R = 1 &- \sum_{c=1}^{K_s}[\prod_{t=1}^{n'} P(\eta_{c,t})] \\
&= 1 - [P(\bar{1}^* \cdot \bar{2}^*) + P(1^* \cdot 2^* \cdot \bar{4} \cdot \bar{5}) + P(\bar{1}^* \cdot 2^* \cdot \bar{3}^* \cdot \bar{5}) + P(\bar{1}^* \cdot 2^* \cdot 3^* \cdot \bar{4} \cdot \bar{5})] + P(1^* \cdot \bar{2}^* \cdot \bar{3}^* \cdot \bar{4}) + P(1^* \cdot \bar{2}^* \cdot 3^* \cdot \bar{4} \cdot \bar{5}) \\
&= 0.7972
\end{aligned}
\tag{47}
$$

### Result analysis

As shown in Fig 9, through the calculation results of the two methods, it can be found that after considering the nodes such as elbows and tees, the reliability of the network system is reduced by about 18%, and the difference is obvious. Therefore, the recursive decomposition algorithm considering nodes needs to be considered for pipeline design or post-earthquake rescue, as long as it involves the reliability calculation of the pipeline network.

## Calculation example: Large complex pipeline network

### Pipeline network condition

The reliability of the pipeline is affected by the pipeline itself and the soil environment in which it is located, including the pipeline material, the type of soil, and the physical properties

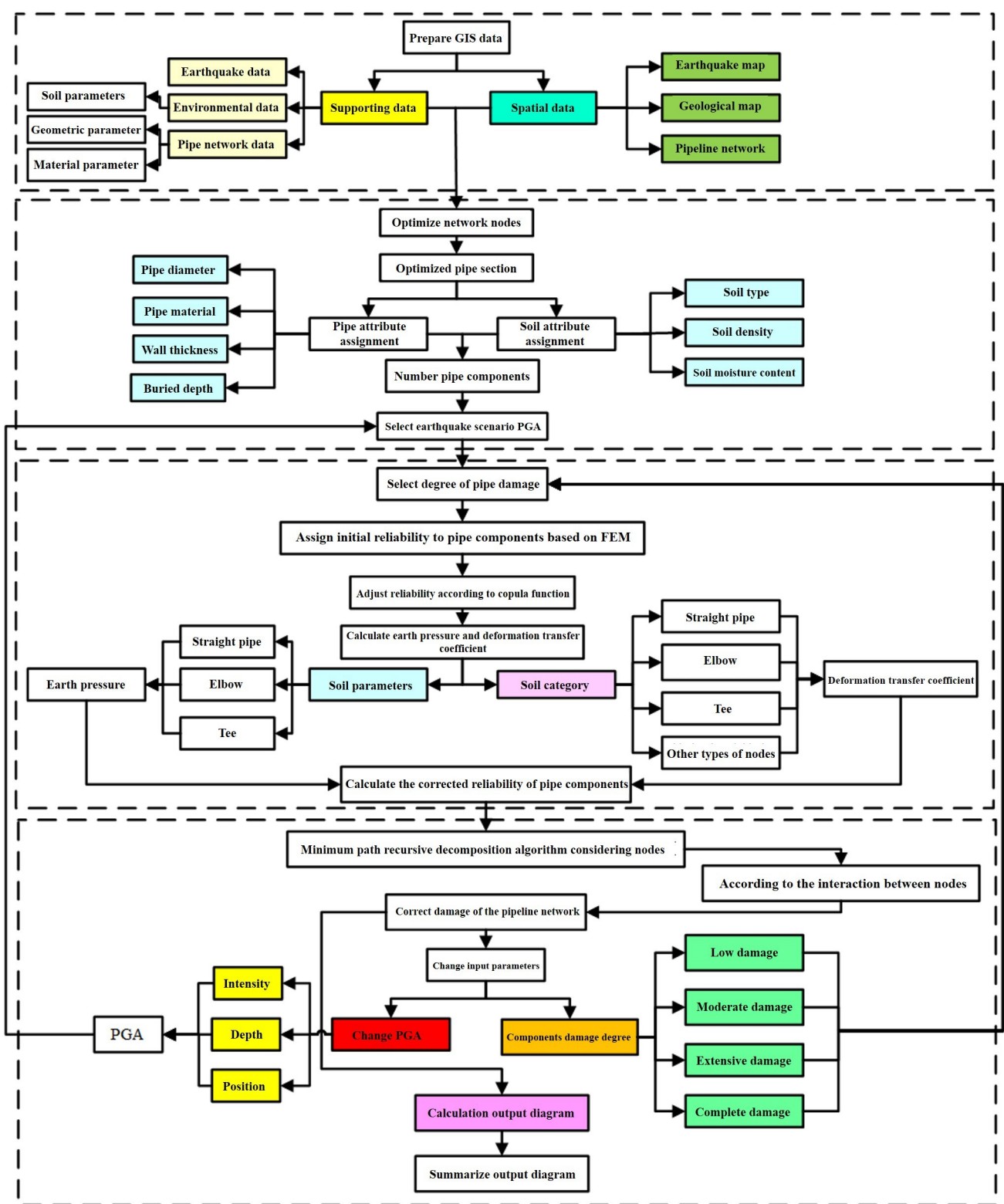

**Fig 10. The calculation process of pipeline network reliability.**

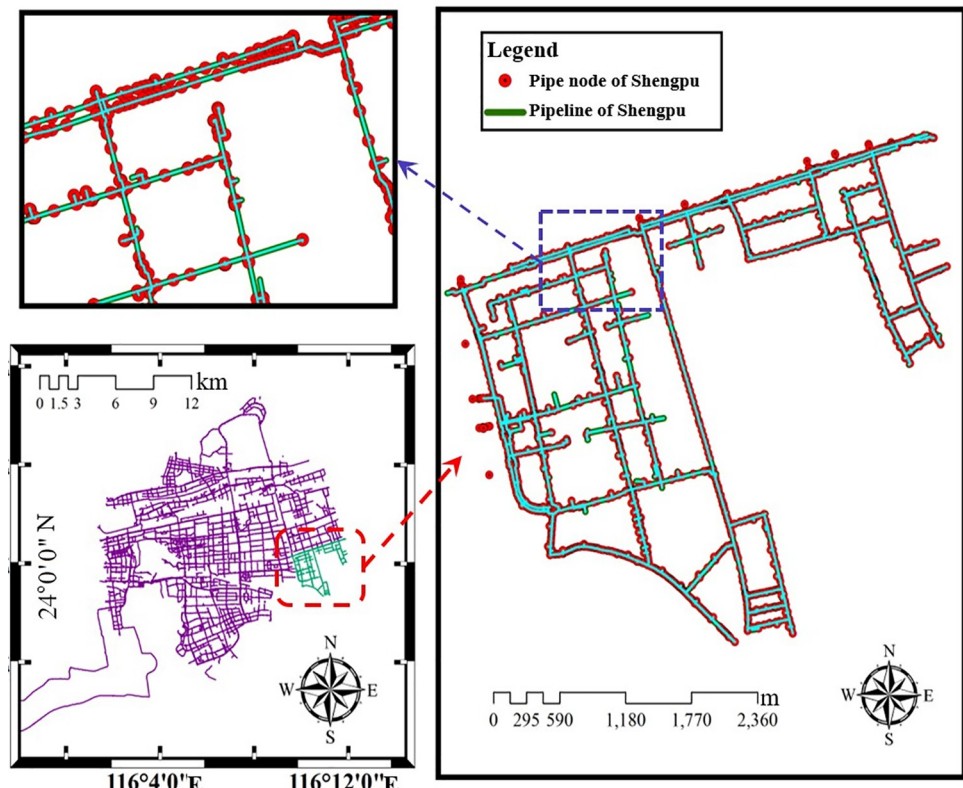

**Fig 11. Nodes of water supply network in Shengpu subregion of Suzhou Industrial Park.** (Republished under a CC BY license, with permission from Suzhou Industrial Park Qingyuan Huayan Water Co. LTD, original copyright 2019).

of the soil. In this paper, two factors including the earth pressure and the pipe-soil deformation transfer coefficient were considered as the influencing factors of the reliability of the pipe section. Among them, properties of the soil, the pipe diameter, and burial depth were considered for the earth pressure; the seismic wave wavelength, the pipe elastic modulus, the pipe cross-sectional area, the spring coefficient of the soil, and the angle of seismic wave incidence were considered for the straight pipe deformation transfer coefficient; the pipe-soil friction coefficient, the pipe wall thickness, the pipe diameter, the pipe wall thickness, and the angle of the elbow were considered for bends. Based on the pipe itself and the external environment, the influence of the change of earth pressure and deformation transfer coefficient on the change of reliability of the pipe section was taken to realize the adjustment of the reliability of the same type of pipe section. The functions of the pipeline network connectivity reliability calculation system mainly include four modules, and its main functions and analysis flow are shown in Fig 10.

A large-scale water supply pipeline network with complex topology was used to calculate its connectivity reliability in the Shengpu subregion of Suzhou Industrial Park in this example. The above reliability calculation results of various types of pipes and nodes are used in the calculation process. Based on the geographic information system (GIS) software platform, by referring to the function and module design of the developed intelligent supervision platform for the safe operation of pipeline networks, combined with the requirements of the operation safety supervision of municipal pipeline network in the park, this study adopts a unified database format and considers the node reliability to carry out the design of GIS background

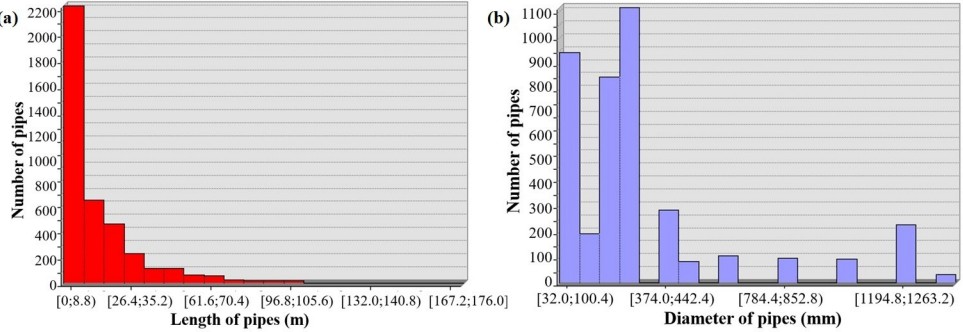

**Fig 12. Length and diameter distribution of water supply pipelines in Shengpu subregion.** (a) Length, (b) Diameter.

calculation plug-in for pipeline network connectivity. At the same time, for the safety requirements, the water supply pipeline network focuses on the safety risk assessment of the pipeline network operation, the intelligent analysis of the leakage of the pipeline network, the simulation model of the pipeline network operation, and sensing system of pipeline safety monitoring [36–38].

The water supply network of the whole Suzhou Industrial Park is huge. Although the area of the Shengpu subregion is only 8 square kilometers, accounting for less than 5% of the area of Suzhou Industrial Park, 3850 pipe sections and 2420 pipe connection nodes are involved. As shown in Fig 11, it can also be seen from the enlarged view of the pipe section that the density of pipeline connection nodes is large. Although there are many kinds of pipe lengths and pipe diameters, their lengths are concentrated in the range of 0~20 m, and the pipe diameters are all concentrated in the range of 50~350 mm. The specific distribution is shown in Fig 12.

Fig 13 shows the location of different types of pipes for the water supply network in the Shengpu subregion, including different pipe lengths, materials, diameters, and construction methods. It can be found that long pipes and large diameter ductile iron pipes are mainly used in the mainline, and the construction method is immersed pipe; the PE pipes are also used for some pipelines, and the construction method is direct excavation.

Since there are more than 2000 nodes in the Shenpu Partition, which brings difficulties to the connectivity calculation. Therefore, this section only selects the nodes that have a greater impact on the calculation results as the calculation nodes, such as elbows, tees, crosses, and other important nodes. There are 349 necessary nodes in total. The pipe sections with non-essential nodes are connected to synthesized 326 pipe sections. And the nodes and pipe sections are numbered in turn, as shown in Fig 14.

## Calculation results

Based on Table 4 and the reliability of every pipe section not listed, the general weighted network non-intersecting minimum path recursive decomposition algorithm with not considering nodes and considering elbow and tee nodes is used to calculate the seismic connectivity reliability of the large complex pipeline network respectively. The connectivity calculation result without considering nodes is shown in Fig 15, and the connectivity calculation result considering nodes is shown in Fig 16.

## Result analysis

It can be seen from Fig 16 that the damage degree of the pipe section without considering nodes is smaller than that of considering nodes. The failure of a node usually leads to the

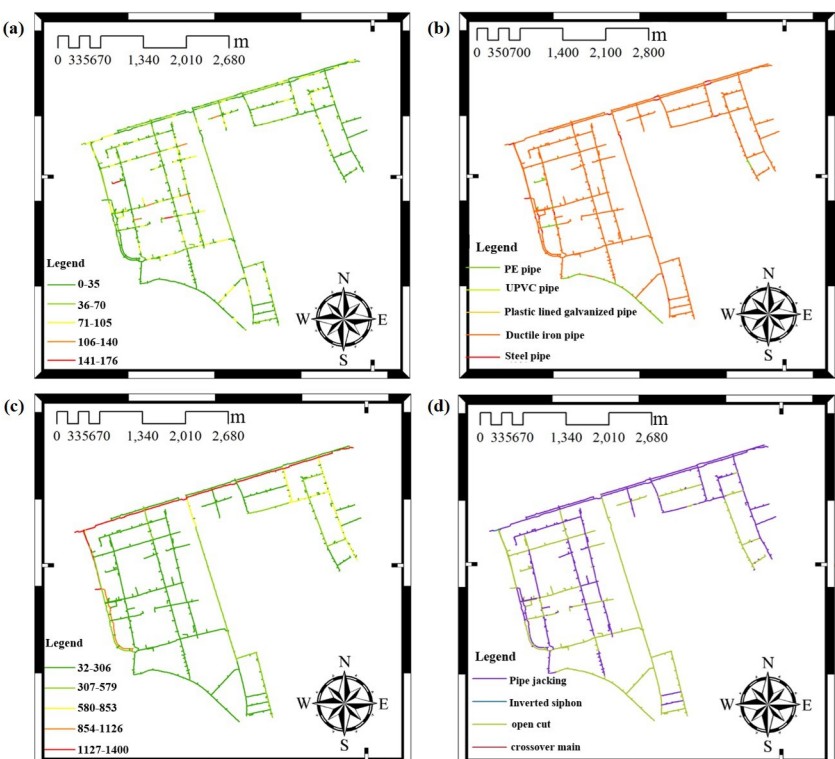

**Fig 13. Classification of different types of water supply pipelines in Shengpu subregion.** (a) Length, (b) Material, (c) Diameter, (d) Construction method. (Map image courtesy of Suzhou Industrial Park Qingyuan Huayan Water Co. LTD).

failure of the whole pipe sections of the path. The probability of failure is greater in the area with dense elbow and tee nodes. The damage without considering nodes mostly occurs at the connection of complex pipe sections.

Fig 17 shows the calculation results of the four damage levels of low, moderate, extensive, and complete damage loaded with the case of the peak ground acceleration of 0.2*g* for the pipeline network in the Shengpu subregion. It can be found that the higher the damage level is, the

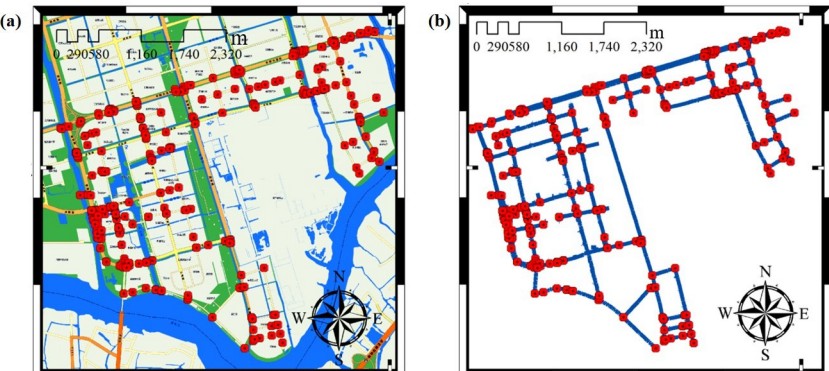

**Fig 14. Necessary nodes and topological diagram of pipeline network after simplification.** (a) Necessary nodes, (b) Pipeline network topology. (Republished under a CC BY license, with permission from Suzhou Industrial Park Qingyuan Huayan Water Co. LTD, original copyright 2019).

**Table 4. Probability of low damage to pipe section in Shengpu Road loaded with PGA = 0.2*g*.**

| Number of pipe sections or nodes | Road | Pipe component type | Reliability of low damage |
|---|---|---|---|
| P-151 | Shengpu Road (Jianpu Street) | Pipe | 0.963 |
| T-82 | Shengpu Road (Jianpu Street) | Tee | 0.969 |
| P-152 | Shengpu Road (Jianpu Street) | Pipe | 0.963 |
| T-83 | Shengpu Road (Jianpu Street) | Tee | 0.969 |
| P-153 | Shengpu Road (Jianpu Street) | Pipe | 0.963 |
| T-84 | Shengpu Road (Jianpu Street) | Tee | 0.969 |
| P-154 | Shengpu Road (Jianpu Street) | Pipe | 0.907 |
| E-74 | Shengpu Road (Jianpu Street) | Elbow | 0.430 |
| P-155 | Shengpu Road (Jianpu Street) | Pipe | 0.907 |
| E-75 | Shengpu Road (Jianpu Street) | Elbow | 0.430 |
| P-156 | Shengpu Road (Jianpu Street) | Pipe | 0.907 |
| E-76 | Shengpu Road (Jianpu Street) | Elbow | 0.430 |
| P-157 | Shengpu Road (Jianpu Street) | Pipe | 0.907 |
| E-77 | Shengpu Road (Jianpu Street) | Elbow | 0.430 |
| P-158 | Shengpu Road (Jianpu Street) | Pipe | 0.907 |
| E-78 | Shengpu Road (Jianpu Street) | Elbow | 0.430 |
| P-159 | Shengpu Road (Jianpu Street) | Pipe | 0.907 |
| E-79 | Shengpu Road (Jianpu Street) | Elbow | 0.430 |
| P-160 | Shengpu Road (Jianpu Street) | Pipe | 0.963 |
| T-85 | Shengpu Road (Jianpu Street) | Tee | 0.969 |
| P-161 | Shengpu Road (Jianpu Street) | Pipe | 0.963 |
| T-86 | Shengpu Road (Jianpu Street) | Tee | 0.969 |
| P-162 | Shengpu Road (Jianpu Street) | Pipe | 0.963 |
| T-87 | Shengpu Road (Jianpu Street) | Tee | 0.969 |
| P-163 | Shengpu Road (Jianpu Street) | Pipe | 0.907 |
| E-80 | Shengpu Road (Jianpu Street) | Elbow | 0.430 |
| P-164 | Shengpu Road (Jianpu Street) | Pipe | 0.907 |
| E-81 | Shengpu Road (Jianpu Street) | Elbow | 0.430 |
| P-165 | Shengpu Road (Jianpu Street) | Pipe | 0.907 |
| E-82 | Shengpu Road (Jianpu Street) | Elbow | 0.430 |
| P-166 | Shengpu Road (Jianpu Street) | Pipe | 0.963 |
| T-88 | Shengpu Road (Jianpu Street) | Tee | 0.969 |
| P-167 | Shengpu Road (Jianpu Street) | Pipe | 0.963 |
| T-89 | Shengpu Road (Jianpu Street) | Tee | 0.969 |
| P-168 | Shengpu Road (Jianpu Street) | Pipe | 0.963 |
| T-90 | Shengpu Road (Jianpu Street) | Tee | 0.969 |
| P-169 | Shengpu Road (Jianpu Street) | Pipe | 0.963 |

less likely the pipe section is to be damaged, i.e., the less the failure probability of the pipe section is.

Table 5 shows the probability of failure probability greater than 50% based on these four kinds of damage considering the elbow, tee, and other node. It can be seen from the table that the possibility of extensive damage to the pipeline reaches 9.02%, while the possibility of extensive damage to the node reaches 12.29%. At this time, the pipeline network can not supply water normally.

The existence of nodes such as elbows and tees leads to a reduction in the connectivity reliability of the pipeline network, but the damage at the nodes is often caused by connection

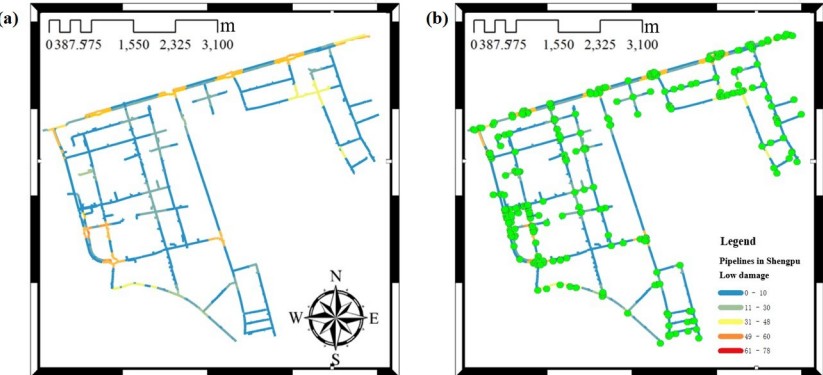

**Fig 15. Low damage results from the connectivity to the water supply pipeline network in Shengpu subregion without considering the nodes.** (a) Not considering the nodes, (b) Not considering the nodes (including the nodes). (Republished under a CC BY license, with permission from Suzhou Industrial Park Qingyuan Huayan Water Co. LTD, original copyright 2019).

damage, so the external dynamic effects can be resisted by improving the strength of the connection between node components and straight pipes [39]; adding special buffer layers at the contact between nodes and the soil can also effectively reduce the force of the soil on the pipe nodes [40]; in the design process of the pipeline network, making the pipe nodes avoid fault zones or areas prone to liquefaction. After the combination of pre-design and construction treatment, the seismic reliability of the nodes can be effectively improved.

## Discussion

The network analytic hierarchy process (ANP) was proposed by Professor Satty and developed based on the analytic hierarchy process (AHP) [41]. ANP is composed of a control layer and network layer, and the elements in ANP are interdependent and dominate each other to form a network structure, as shown in Fig 18. Considering the coupling factor weight and multi-factor comprehensive effect [42], the fuzzy comprehensive evaluation method based on ANP is used in this study, as shown in Fig 19(A). The calculation results are compared with the connection reliability of the pipeline network under the above earthquake. During the calculation,

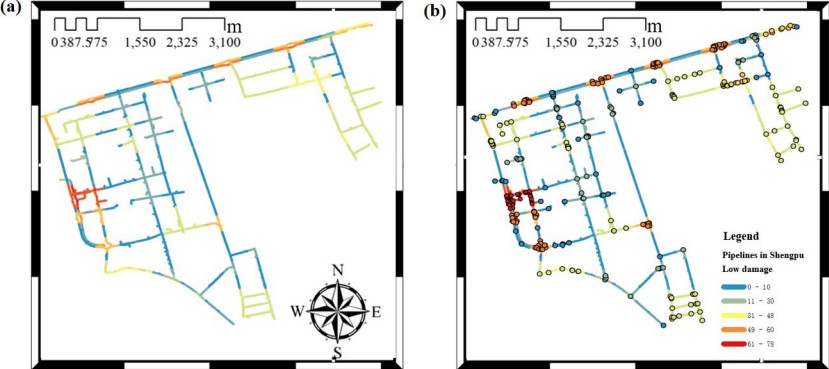

**Fig 16. Low damage results from the connectivity to the water supply pipeline network in Shengpu subregion considering the nodes.** (a) Considering the nodes, (b) Considering the nodes (including the nodes). (Republished under a CC BY license, with permission from Suzhou Industrial Park Qingyuan Huayan Water Co. LTD, original copyright 2019).

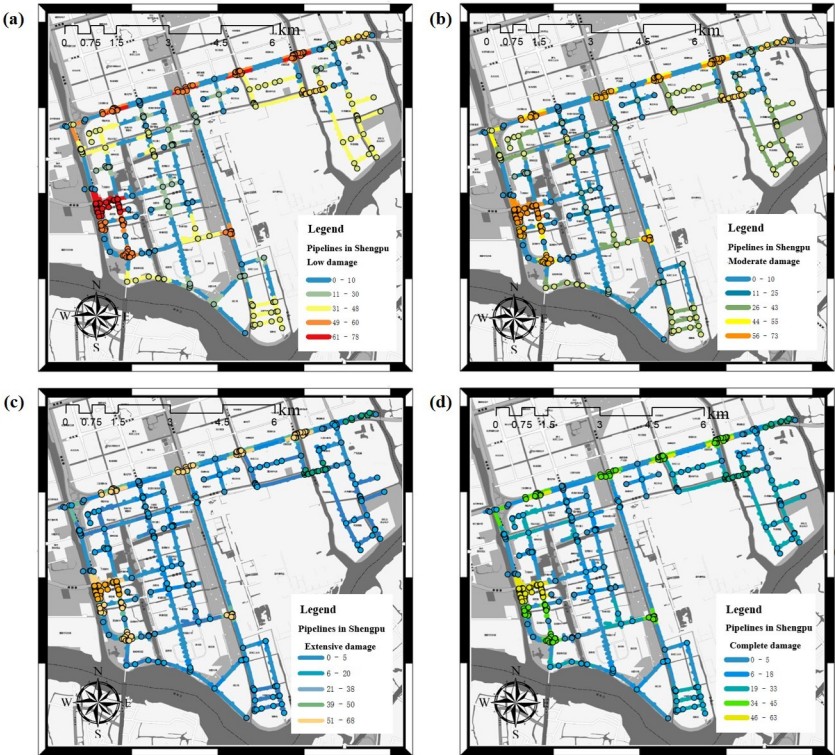

**Fig 17. Failure probability of four damage levels to the pipeline network (PGA = 0.2g).** (a) Low damage, (b) Moderate damage, (c) Extensive damage, (d) Complete damage. (Republished under a CC BY license, with permission from Suzhou Industrial Park Qingyuan Huayan Water Co. LTD, original copyright 2019).

the fragility $G$ of the pipe is determined by the vulnerability index $V$, risk cause index $R$ and risk consequence index $\gamma$ of the pipe, as shown in Eqs (48)–(50).

$$G = (V + R) \cdot \gamma \tag{48}$$

$$V = \sum_{i=1}^{n} (V_i \cdot v_i) \tag{49}$$

$$R = T \cdot r_1 + C \cdot r_2 + I \cdot r_3 + \cdots + N \cdot r_m \tag{50}$$

The specific construction process of the ANP method is as follows [43]:

(1) Identifying risk factors and building ANP networks

To be consistent with the factors considered in the above numerical simulations, according to the research results, the cause of the pipeline accident is set to be controlled by the earthquake disaster, form, material and geometric characteristics of the pipe components, and physical properties of the soil environment surrounding the pipe. Establish risk index criteria $U(1)$,

**Table 5. Seismic connectivity reliability of water supply network in Shengpu subregion.**

| Damage degree | More than 50% for failure probability | | | |
|---|---|---|---|---|
| | Number of pipes | Percentage (%) | Number of nodes | Percentage (%) |
| Low | 58 | 18.13 | 126 | 36.10 |
| Moderate | 51 | 15.74 | 59 | 17.14 |
| Extensive | 29 | 9.02 | 42 | 12.29 |
| Complete | 20 | 6.13 | 25 | 7.21 |

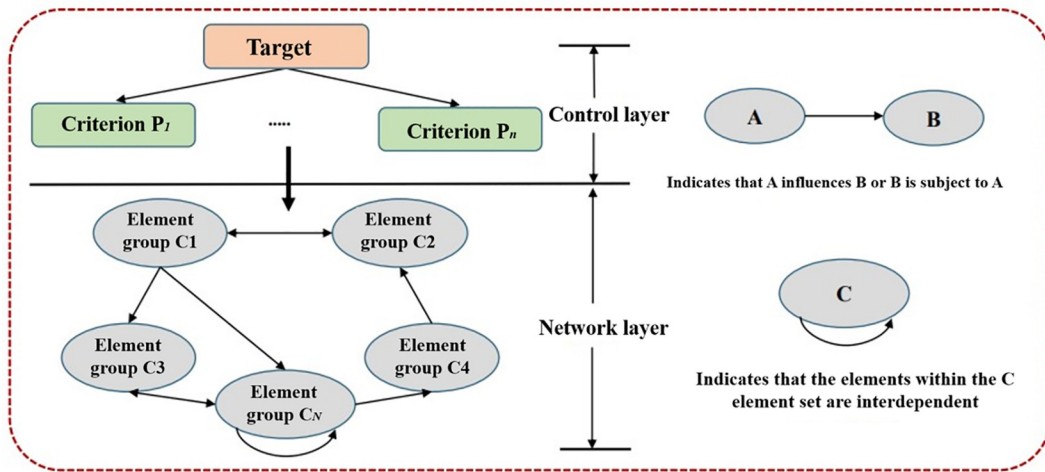

**Fig 18. Typical ANP structure diagram.**

$U(2)$ and $U(3)$, representing the effects of earthquakes, the nature of the pipeline itself, and the nature of the soil, respectively.

(2) Establishing a judgment matrix

The three-level structure system is used to establish the judgment matrix from the standard level to the target level. For example, the three-level index $U(2,1,1)$ is used as the judgment standard, and $U(2,1)$, $U(2,2)$ and $U(2,3)$ are used to obtain the judgment matrix, where $U(2,2)$ represents the influence factors of the elbow, specifically including diameter, wall thickness, elbow angle and material, and the judgment matrix is shown in Eq (51).

$$A_{U(2,2)-U(2,1,1)} = \begin{pmatrix} a_{11} & a_{12} & a_{13} & a_{14} \\ a_{21} & a_{22} & a_{23} & a_{24} \\ a_{31} & a_{32} & a_{33} & a_{34} \\ a_{41} & a_{42} & a_{43} & a_{44} \end{pmatrix} \tag{51}$$

where $a_{ij}$ is assigned a value based on the relative sizes of $i$ and $j$, and then the judgment matrix is normalized.

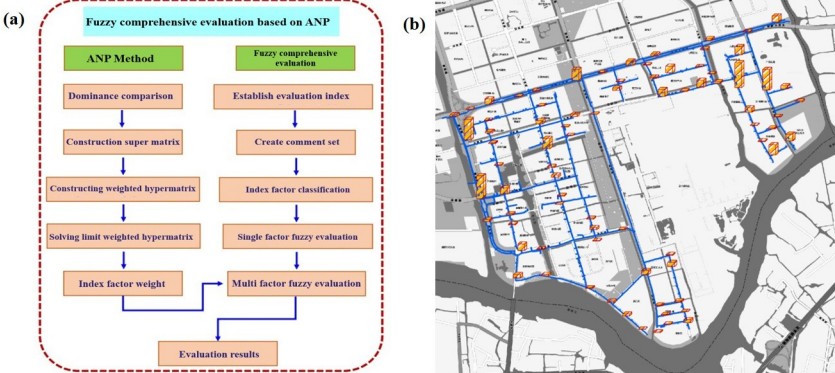

**Fig 19. Fuzzy comprehensive evaluation method based on ANP.** (a) Calculation flow chart, (b) Risk calculation results of pipeline network in Shengpu subregion. (Republished under a CC BY license, with permission from Suzhou Industrial Park Qingyuan Huayan Water Co. LTD, original copyright 2019).

(3) Establishing a supermatrix

Using all three-level indicators in $U(2)$ as judgment criteria, a judgment matrix is constructed and normalized to integrate the initial supermatrix as follows.

$$W_{S,U(2)} = \begin{pmatrix} w_{11} & w_{12} & w_{13} \\ w_{21} & w_{22} & w_{23} \\ w_{31} & w_{32} & w_{33} \end{pmatrix} \tag{52}$$

(4) Establishing a weighted supermatrix

Taking $U(2,1)$, $U(2,2)$ and $U(2,3)$ as the evaluation criteria of the importance of the second-level indicators, the second-level indicators of $U(2)$ are compared and judged to obtain the judgment matrix. Then normalization is performed to get the weight matrix $B_{U(2)}$. The weighted supermatrix is obtained by multiplying the elements of the initial supermatrix and the corresponding position elements of the weight matrix.

$$\bar{W}_{S,U(2)} = W_{S,U(2)} \times B_{U(2)} = \begin{pmatrix} b_{11} \cdot w_{11} & b_{12} \cdot w_{12} & b_{13} \cdot w_{13} \\ b_{21} \cdot w_{21} & b_{22} \cdot w_{22} & b_{23} \cdot w_{23} \\ b_{31} \cdot w_{31} & b_{32} \cdot w_{32} & b_{33} \cdot w_{33} \end{pmatrix} \tag{53}$$

(5) Indicator weights based on supermatrix limits

Each tertiary indicator in $U(2)$ can be self-multiplied infinitely to obtain a weighted supermatrix until the matrix converges, as shown below:

$$\bar{W}_{S,U(2)} = \lim_{n \to \infty} \left( \bar{W}_{S,U(2)} \right)^n \tag{54}$$

Determine the classification criteria and risk level of each factor of $U(1)$, $U(2)$ and $U(3)$. Based on the above ANP calculation process and the classification of risk levels, the risk analysis of the water supply network of Shengpu subregion is carried out. The calculation results are shown in the histogram of Fig 19(B). The height of the column at each position represents the degree of risk. Combined with Fig 16, it can be found that the damage risk is serious at the nodes, and the distribution area with high damage risk value is highly consistent with the area with high connectivity failure probability calculated by the probability analysis algorithm. Combined with Fig 10, it also reflects that the dynamic failure of the pipeline is closely related to the attributes of the pipeline itself and the soil environment in which the pipeline is located. Table 6 shows the number of pipe sections and nodes with a risk of pipeline damage greater than 50% and their corresponding percentages calculated based on the ANP method. It can be found that the pipeline damage calculated based on the ANP method is smaller than that of the minimum path recursive decomposition algorithm considering nodes proposed, but the relative deviations of both are within 20%. Therefore, the calculation results of the minimum path recursive decomposition algorithm considering nodes can be considered to be in line with the actual situation.

## Conclusions

Based on the accuracy of the calculation method of the pipeline network connectivity, a calculation method for large and complex pipeline network connectivity considering nodes was proposed. Taking the simple pipeline network and the complex water supply network in

**Table 6. Risk degree analysis of water supply network in Shengpu subregion based on ANP method.**

| Damage degree | More than 50% for failure risk probability | | | | | |
|---|---|---|---|---|---|---|
| | Number of pipes | Percentage (%) | Relative deviation (%) | Number of nodes | Percentage (%) | Relative deviation (%) |
| Low | 49 | 15.03 | 17.10 | 117 | 33.52 | 7.15 |
| Moderate | 45 | 13.80 | 12.33 | 48 | 13.75 | 19.78 |
| Extensive | 26 | 7.98 | 11.53 | 37 | 10.60 | 13.75 |
| Complete | 18 | 5.52 | 9.95 | 23 | 6.59 | 8.60 |

Suzhou Industrial Park as examples, the impact of the damage of elbow and tee nodes on the connectivity of the pipeline network is elaborated, and the specific conclusions drawn are as follows:

1. For the connectivity calculation of the pipeline network, the reliability of nodes is embedded in the reliability of edges, and a more detailed and comprehensive minimum path recursive decomposition algorithm considering elbow, tee, and other nodes is deduced based on the general minimum path recursive decomposition algorithm.

2. Based on the reliability calculation theory of pipe sections and nodes, the reliability of the straight pipe, elbow, and tee for different soil properties and four different damage degrees is obtained by using the finite element method.

3. After considering the nodes such as elbows and tees, the reliability of the network system is significantly reduced, and the failure of one node usually causes the failure of the pipe sections of the path. The probability of failure is higher in the area with dense elbow and tee nodes, and the damage without considering the nodes mostly occurs at the connecting positions of complex pipe segments.

4. Taking the water supply pipeline network in the Shengpu subregion of Suzhou Industrial Park as an example, when the PGA is 0.2$g$, the network connectivity reliability is calculated through the minimum path recursive decomposition algorithm considering the nodes. The result is that the possibility of extensive damage to the pipe section is 9.02%, while the possibility of extensive damage to the node is 12.29%. At this time, the pipeline can not supply water normally. The relative deviations of the calculation results obtained by the proposed method and the ANP risk analysis method are within 20%, which are of certain guiding significance.

5. Although the minimum path recursive decomposition algorithm considering elbow, tee, and other nodes does not solve the problem of computing speed of large-scale complex networks, it makes a detailed consideration of the calculation of elbow nodes, tee nodes, and other types of nodes that are more likely to be damaged. Therefore, the calculation accuracy of the algorithm has been improved to a certain extent, which lays a foundation for the further accurate calculation of pipeline network connectivity.

## Supporting information

**S1 Table. Probability of low damage of each pipe section based on FEM.**
(XLSX)

**S2 Table. Probability of moderate damage of each pipe section based on FEM).**
(XLSX)

**S3 Table. Probability of extensive damage of each pipe section based on FEM.**
(XLSX)

**S4 Table. Probability of complete damage of each pipe section based on FEM.**
(XLSX)

# Author Contributions

**Conceptualization:** Delong Huang.

**Data curation:** Delong Huang.

**Funding acquisition:** Zhongling Zong.

**Investigation:** Aiping Tang.

**Methodology:** Delong Huang, Zhongling Zong.

**Project administration:** Zhongling Zong, Aiping Tang.

**Supervision:** Aiping Tang.

**Visualization:** Zhongling Zong.

**Writing – original draft:** Delong Huang.

**Writing – review & editing:** Aiping Tang.

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
