## [Decision Letter · Decision Letter 0]

7 Jun 2022

PONE-D-22-14650Study on Connectivity of Buried Pipeline Network Considering Nodes Reliability under Seismic ActionPLOS ONE

Dear Dr. Huang,

Thank you for submitting your manuscript to PLOS ONE. After careful consideration, we feel that it has merit but does not fully meet PLOS ONE’s publication criteria as it currently stands. Therefore, we invite you to submit a revised version of the manuscript that addresses the points raised during the review process.

Please consider all comments

We look forward to receiving your revised manuscript.

Kind regards,

Ahmed Mancy Mosa, Ph.D.

Academic Editor

PLOS ONE

Journal Requirements:

"This research was supported by the Hainan Province Key R&D Program (Social Development) Project of China (No. ZDYF2022SHFZ089), and the Jiangsu Province Key R&D Program (Social Development) Project of China (No. BE2021681)."

"This research was supported by the Hainan Province Key R&D Program (Social Development) Project of China (No. ZDYF2022SHFZ089), and the Jiangsu Province Key R&D Program (Social Development) Project of China (No. BE2021681)."

5. We note that Figures 2, 10, 12, 13, 14, 15, 16, and 18 in your submission contain [map/satellite] images which may be copyrighted. All PLOS content is published under the Creative Commons Attribution License (CC BY 4.0), which means that the manuscript, images, and Supporting Information files will be freely available online, and any third party is permitted to access, download, copy, distribute, and use these materials in any way, even commercially, with proper attribution. For these reasons, we cannot publish previously copyrighted maps or satellite images created using proprietary data, such as Google software (Google Maps, Street View, and Earth). For more information, see our copyright guidelines: http://journals.plos.org/plosone/s/licenses-and-copyright.

a. You may seek permission from the original copyright holder of Figures 2, 10, 12, 13, 14, 15, 16, and 18 to publish the content specifically under the CC BY 4.0 license.  

Reviewers' comments:

Reviewer's Responses to Questions

**Comments to the Author**

1. Is the manuscript technically sound, and do the data support the conclusions?

Reviewer #1: Yes

Reviewer #2: Yes

2. Has the statistical analysis been performed appropriately and rigorously? 

Reviewer #1: Yes

Reviewer #2: Yes

3. Have the authors made all data underlying the findings in their manuscript fully available?

Reviewer #1: Yes

Reviewer #2: Yes

4. Is the manuscript presented in an intelligible fashion and written in standard English?

Reviewer #1: Yes

Reviewer #2: Yes

5. Review Comments to the Author

Reviewer #1: The article entitled ‘Study on Connectivity of Buried Pipeline Network Considering Nodes Reliability under Seismic Action’ studies the reliability of the connectivity of complex pipeline networks of nodes of pipelines.

The paper is well organized and structured. However, the English is rather poor. It is suggested to be reviewed by a native speaker. Additionally, there are some grammatical and typographical errors. In some parts, it is very difficult to understand and read what they are proposing. The abstract should be re-written avoiding long sentences and clearly indicating the novelty and the objective of this work.

My main concern is the lack of novelty of this work. The authors do not really highlight what they have done compared to other works. Also, the list of references is scarce and it is not updated. Therefore, it is very complicated to understand why this work is worth being published.

Abstract#

It should be re-written.

Introduction#

Figure 1 is not clear nor explained. Also, it is not referenced in the text.

Results and conclusions#

The results part is rather scarce. Additional comments could be added to analyse the results and to ultimately improve the quality of the paper. Conclusions could be improved after enhancing the results part.

Reviewer #2: The paper presents an improved method to assess reliability of connectivity of buried pipeline network considering nodes, deduced a more detailed and comprehensive non-intersecting minimum path recursive decomposition algorithm considering nodes such as elbows and tees. The study result shows that after considering the elbow, tee, and other nodes, the reliability of the network system decreases significantly. The failure of one node usually leads to the failure of the whole pipeline section, and the failure probability is greater in the area with dense elbow and tee nodes.

（1） As shown in Fig.3, the elbow and tee are buried in two type soils, but the properties of the two type soils are not given. How to taking into account of the influence of different soils to elbow and tee under earthquake?

（2） Apart from part 3, in what else way can the method be further testified?

（3） In part 5” …based on the famous 472 analytic hierarchy process…”, the “famous” is not needed.

（4） The top left small figure of Fig.10 is an enlarged local of the right figure, so a circle is needed for right figure indicating the position of the local area.

（5） The study result shows that the nodes will significantly reduce the reliability of the network. Can the authors give some suggestions to improve the node reliability?

In general, the paper improved reliability analysis method by considering nodes, and the study result becomes more realistic. This new method can be further used in pipeline network analysis. The paper will be ready for publication after modification.

6. PLOS authors have the option to publish the peer review history of their article (what does this mean?). If published, this will include your full peer review and any attached files.

Reviewer #1: No

Reviewer #2: No

---

## [Author Response · Author response to Decision Letter 0]

18 Jun 2022

Editor’s Comments:

Response: Thank you for your comment. We have typeset the manuscript as required by the template, and we have also uploaded the manuscript file with the specified name and the comment response file as required.

Response: Thank you for your comment. The results listed in the manuscript are implemented exactly according to the formulae proposed in the manuscript and the calculation steps, so no software code is involved.

3. Please remove any funding-related text from the manuscript and let us know how you would like to update your Funding Statement.

Response: We are very sorry for our negligence of the Funding Statement. We have remove funding-related text from the manuscript.

4. Please ensure that you have an ORCID iD and that it is validated in Editorial Manager.

Response: Thank you for your comment. I have updated my information in the system about ORCID iD.

5. We note that Figures 2, 10, 12, 13, 14, 15, 16, and 18 in your submission contain [map/satellite] images which may be copyrighted. All PLOS content is published under the Creative Commons Attribution License (CC BY 4.0), which means that the manuscript, images, and Supporting Information files will be freely available online, and any third party is permitted to access, download, copy, distribute, and use these materials in any way, even commercially, with proper attribution. For these reasons, we cannot publish previously copyrighted maps or satellite images created using proprietary data, such as Google software (Google Maps, Street View, and Earth). For more information, see our copyright guidelines: http://journals.plos.org/plosone/s/licenses-and-copyright.

We require you to either (1) present written permission from the copyright holder to publish these figures specifically under the CC BY 4.0 license, or (2) remove the figures from your submission.

Response: Thank you for your comment. This article is based on the research projects "the Hainan Province Key R&D Program (Social Development) Project of China (No. ZDYF2022SHFZ089), and the Jiangsu Province Key R&D Program (Social Development) Project of China (No. BE2021681)". The projects required that the method we proposed should have demonstration projects to test the feasibility of the method, so the project team cooperated with Suzhou Industrial Park Qingyuan Huayan Water Co. LTD to provide us with the pipe network data. The map information in Figures 2, 10, 12, 13, 14, 15, 16, and 18 (modified figure numbers: Figure 2, 11, 13, 14, 15, 16, 17, 19 in the revised manuscript), which are covered in this manuscript, were obtained from Suzhou Industrial Park Qingyuan Huayan Water Co. LTD, which owns the rights to use and publish the map information. Therefore, this manuscript has added the caption in the captions of Figures 2, 10, 12, 13, 14, 15, 16, and 18 (modified figures numbered: Figures 2, 11, 13, 14, 15, 16, 17, 19 in the revised manuscript): Republished under a CC BY license, with permission from Suzhou Industrial Park Qingyuan Huayan Water Co. LTD, original copyright 2019. We have provided the Permission form in the form of Supporting files. And we have added "Data Availability Statement" at the end of the manuscript. Please contact me if you need any other information about this content, thank you!

Reviewers' comments:

Reviewer #1

Comment 1:

1. The paper is well organized and structured. However, the English is rather poor. It is suggested to be reviewed by a native speaker. Additionally, there are some grammatical and typographical errors. In some parts, it is very difficult to understand and read what they are proposing. The abstract should be re-written avoiding long sentences and clearly indicating the novelty and the objective of this work.

Response: First of all, thank you very much for your comments. We have had a native English-speaking editor revise the language of the manuscript in depth, including grammar and sentence structure, and the results are detailed in "Revised Manuscript with Track Changes". This manuscript has been reformatted in accordance with the journal's formatting requirements. We have also rewritten the abstract of this manuscript and reorganized the problems studied in this manuscript to express the novelty of this work. Long sentences have been replaced by multiple short sentences with synonyms. Please see the abstract section of the manuscript for details of the revisions.

2. My main concern is the lack of novelty of this work. The authors do not really highlight what they have done compared to other works. Also, the list of references is scarce and it is not updated. Therefore, it is very complicated to understand why this work is worth being published.

Response: Thank you for your comment. The innovative point of this paper is to propose a probabilistic analysis algorithm of pipeline network connectivity considering nodes, which enables the accuracy of the calculation results of pipeline network connectivity under earthquake effects to be effectively improved. It provides an accurate basis for the seismic design and post-disaster rescue of pipeline networks. We updated the cited references and cited more academic achievements related to this study in recent three years. The research results of this paper are of great significance to the seismic connectivity calculation and risk assessment of buried pipeline network.

3. Abstract#

It should be re-written.

Response: Thank you very much for your comments. By sorting out the innovative points of the article, we have rewritten the abstract. Please see the revised manuscript for the detailed revision results.

4. Introduction#

Figure 1 is not clear nor explained. Also, it is not referenced in the text.

Response: We are very sorry for our negligence of Figure 1. We have added a reference and an explanation to Figure 1 in lines 40-43.

5. Results and conclusions#

The results part is rather scarce. Additional comments could be added to analyse the results and to ultimately improve the quality of the paper. Conclusions could be improved after enhancing the results part.

Response: First of all, thank you very much for your comments. This manuscript adds measures to improve the reliability of pipeline nodes in the results analysis section for large pipeline networks. In the discussion section, the calculation process of ANP method for risk assessment of pipeline networks and the calculation results in Table 6 are added and compared with the method proposed in this paper. These added discussions can effectively improve the quality of the article. And the description of the ANP method is also added in the conclusion section. Details of the revisions can be viewed in lines 504-510, 524-569, and 585-590.

Reviewer #2

1. As shown in Fig.3, the elbow and tee are buried in two type soils, but the properties of the two type soils are not given. How to taking into account of the influence of different soils to elbow and tee under earthquake?

Response: Thank you very much for your comments. According to the reviewer’s comment, we have added the physical properties of the two soils involved, including test methods, basic physical parameters, and dynamic properties, as detailed in lines 290-301 of the manuscript. And for the effects of different soils on elbows and tees under seismic action, we have mainly used the pipe-soil interaction, including soil pressure and pipe-soil deformation transfer coefficient, and both quantities involve multiple factors in the soil and the pipe itself. Based on these factors, we correct the results calculated by numerical simulation to obtain the reliability of pipe sections at other locations. Please refer to lines 418-430 of the manuscript for details.

2. Apart from part 3, in what else way can the method be further testified?

Response: Thank you for your comment. The calculation results for the large pipeline network have been verified in two ways. One is to calculate the reliability of the pipeline network using the minimum path recursive decomposition algorithm considering nodes and not considering nodes respectively, and it can be found that the algorithm considering nodes is more in line with the actual situation after comparison. Second is to use the ANP method for the risk assessment calculation, and set the seismic effect as the main risk factor. It is found that the calculation results are similar to the calculation method proposed in this paper, and the relative deviation of both methods is within 20%, which again confirms the reliability of the calculation method proposed in this paper. For details, please refer to the "Calculation example: large complex pipeline network" and "Discussion" sections of the manuscript.

3. In part 5” …based on the famous 472 analytic hierarchy process…”, the “famous” is not needed.

Response: We agree with the reviewer’s comment. We have made the revise to this sentence according to the reviewer’s comment in lines 512-514.

4. The top left small figure of Fig.10 is an enlarged local of the right figure, so a circle is needed for right figure indicating the position of the local area.

Response: We agree with the reviewer’s comment. We have made the revise to Fig.10 (modified figure numbered: Figure 11 in the revised manuscript) according to the reviewer’s comment.

5. The study result shows that the nodes will significantly reduce the reliability of the network. Can the authors give some suggestions to improve the node reliability?

Response: Thank you for your comment. We have added recommendations to improve the reliability of the pipeline network nodes, including optimization of the design and construction of the pipeline network, as detailed in lines 504-510 of the manuscript.

---

## [Decision Letter · Decision Letter 1]

4 Jul 2022

Study on connectivity of buried pipeline network considering nodes reliability under seismic action

PONE-D-22-14650R1

Dear Dr. Huang,

We’re pleased to inform you that your manuscript has been judged scientifically suitable for publication and will be formally accepted for publication once it meets all outstanding technical requirements.

Kind regards,

Ahmed Mancy Mosa, Ph.D.

Academic Editor

PLOS ONE

Additional Editor Comments (optional):

Reviewers' comments:

Reviewer's Responses to Questions

**Comments to the Author**

1. If the authors have adequately addressed your comments raised in a previous round of review and you feel that this manuscript is now acceptable for publication, you may indicate that here to bypass the “Comments to the Author” section, enter your conflict of interest statement in the “Confidential to Editor” section, and submit your "Accept" recommendation.

Reviewer #2: All comments have been addressed

2. Is the manuscript technically sound, and do the data support the conclusions?

Reviewer #2: Yes

3. Has the statistical analysis been performed appropriately and rigorously? 

Reviewer #2: Yes

4. Have the authors made all data underlying the findings in their manuscript fully available?

Reviewer #2: Yes

5. Is the manuscript presented in an intelligible fashion and written in standard English?

Reviewer #2: Yes

6. Review Comments to the Author

Reviewer #2: Reviewers' comments are meticulously addressed by the authors. The paper is improved. No comments from our side

7. PLOS authors have the option to publish the peer review history of their article (what does this mean?). If published, this will include your full peer review and any attached files.

Reviewer #2: No

---

## [Editor Report · Acceptance letter]

11 Aug 2022

PONE-D-22-14650R1 

Study on connectivity of buried pipeline network considering nodes reliability under seismic action 

Dear Dr. Huang:

I'm pleased to inform you that your manuscript has been deemed suitable for publication in PLOS ONE. Congratulations! Your manuscript is now with our production department. 

Kind regards, 

on behalf of

Dr. Ahmed Mancy Mosa 

Academic Editor

PLOS ONE